# Adsorption energies on transition metal surfaces: towards an accurate and balanced description

**Rafael B. Araujo** [1] ✉**, Gabriel L. S. Rodrigues**[1]**, Egon Campos dos Santos** [1] **& Lars G. M. Pettersson** [1] ✉

Density functional theory predictions of binding energies and reaction barriers provide invaluable data for analyzing chemical transformations in heterogeneous catalysis. For high accuracy, effects of band structure and coverage, as well as the local bond strength in both covalent and non-covalent interactions, must be reliably described and much focus has been put on improving functionals to this end. Here, we show that a correction from higher-level calculations on small metal clusters can be applied to improve periodic band structure adsorption energies and barriers. We benchmark against 38 reliable experimental covalent and non-covalent adsorption energies and five activation barriers with mean absolute errors of 2.2 kcal mol⁻¹, 2.7 kcal mol⁻¹, and 1.1 kcal mol⁻¹, respectively, which are lower than for functionals widely used and tested for surface science evaluations, such as BEEF-vdW and RPBE.

Density functional theory (DFT) is one of the most employed theoretical approaches to understanding surface-adsorbate interaction strength and reactivity[1,2]. The effectiveness of possible catalysts in heterogeneous and electrocatalysis is, for instance, an application where DFT provides valuable insights. Despite the recent successful history of DFT, there still exist issues regarding its accuracy in describing the interaction strength of adsorbates on transition metal surfaces[3–6]. The construction of reliable reaction energy paths is based on the relative energies of reactants, intermediates and products and without the desirable accuracy, results might lead to wrong conclusions. Considering the importance of DFT for theoretical studies and predictions in catalysis it is thus pertinent to investigate and propose new DFT-based strategies that could result in improved accuracy when computing adsorption energies and transition state barriers on transition metal surfaces.

Aiming to benchmark the performance of the most commonly used functionals, Wellendorff et al.[4] have compiled experimental adsorption energies of common catalytic reactions on transition metal surfaces (the CE39 dataset). Several popular exchange-correlation functionals like LDA, PBEsol, PW91, PBE, RPBE, and BEEF-vdW were evaluated. Overall, BEEF-vdW (dispersion-corrected GGA) was suggested as the best option with an accurate description for

chemisorbed systems, yet with space for improvement where dispersion plays a more important role. Sharada et al.[7] have investigated the accuracy of adsorption energies obtained using the MS2 and SCAN metaGGAs and also the screened hybrid functional HSE06 (2 more reactions were added to the CE39 data set to form the ADS41 data set). BEEF-vdW and MS2 showed the best results, where the metaGGA MS2 yielded a more balanced accuracy between chemisorbed and physisorbed systems. An adaptively weighted sum (SW-R88) of energies from RPBE and optB88-vdW has been proposed, aiming to properly describe both covalent and non-covalent interactions, and shown to result in errors smaller than BEEF-vdW, for instance.[8] In parallel, Minnesota metaGGA functionals were tested against the CE39 data set where, for closed-shell molecular adsorption, the MN15L emerged with the lower error[9]. However, for open-shell systems, BEEF-vdW showed the best performance followed by GAM and RPBE. Later, Mahlberg et al.[10] have demonstrated that the RPBE + D3 performance is as good as BEEF-vdW when the metal-metal and adsorbate-metal interactions below the first metal layer are excluded from the dispersion correction (in this case the D3 semiempirical approach of Grimme[11]).

Although much work has been done, there is still space for improvement. The only method able to deliver low average errors for both chemisorption and physisorption is SW-R88—given that BEEF-

[1]FYSIKUM, AlbaNova University Center, Stockholm University, SE-106 91 Stockholm, Sweden. ✉e-mail: rafaelbna@gmail.com; lgm@fysik.su.se

vdW tends to yield errors in physisorbed systems while the RPBE + D3 tends to overestimate adsorption of chemisorbed systems and these are the two functionals presenting the lowest errors. The SW-R88 comes with the drawback of not having a specific functional form, which could be an issue when computing atomic forces.[8] Hence, the difficulty to find a computational strategy that accurately and reliably describes both strong and weak covalent, as well as dispersion, interactions in heterogeneous and electro-catalysis, remains.

Barrier heights for the dissociation of small molecules on transition metal surfaces were also benchmarked in the work of Sharada et al.[12] There, a database, SBH10, was built containing references from molecular beam scattering, laser-assisted associative desorption and thermal experiments. Further, the accuracy performance of BEEF-vdW, MS2 and HSE06 was investigated against the experimental barriers. For gas-phase reactions, GGA functionals tend to underestimate barriers due to the intrinsic self-interaction error[13,14]. However, Sharada et al.[12] showed that, for reactions in the SBH10 database, the accuracy delivered by the BEEF-vdW functional is superior compared to approaches like MS2 or HSE06. Their findings also indicated that the theoretical description of barriers closely follows the description of final states (chemisorbed states). That is a result of the close resemblance between chemisorbed state and transition states for dissociative reactions. Therefore, functionals presenting high accuracy for adsorption energies would also accurately describe the barriers.

We illustrate the challenge in Fig. 1 where different steps in the decomposition of methanol on Pd(111) and Ni(111) are calculated using three different functionals. For the initial step, where methanol adsorbs weakly through the OH-group, the optB86b-vdW[15–17] and the van der Waals corrected PBE+D3[11,18] provide chemisorption energies in excellent agreement with the experimental data for both Pd and Ni, while PBE without the vdW correction severely underestimates the interaction. Deprotonating the OH-group to give methoxy (OCH$_3$) reverses the situation with PBE now in excellent agreement, while both optB86b-vdW and PBE+D3 severely overestimate the bond-strength. For chemisorbed CO (before forming CO$_2$), neither of these functionals provides a satisfactory estimate of the chemisorption energy. It is well recognized that more advanced, e.g., hybrid functionals exist that alleviate much of these concerns, but their general application to large, extended systems comes at an exceedingly high computational cost.

For small non-periodic systems, quantum chemical (QC) techniques are available that can (in principle) be converged to the desired accuracy. Thus, a corrective scheme, which combines the accuracy of QC for the local chemical bonding and the applicability of DFT to extended periodic systems to capture effects of band structure and coverage, becomes attractive. Such a strategy was recently developed and employed by Alessio et al.[19] who combined DFT calculations on the periodic system with CCSD(T) calculations on smaller cluster models to determine a state-of-the-art theoretical adsorption energy of CO on MgO(001). This work has largely inspired the present study, but transferring this approach to transition metal (TM) clusters is a serious challenge. Hu et al.[20] have shown that this approach can predict the site where CO adsorbs on Cu(111) by using the hybrid functional B3LYP for the correction. Using a small metal cluster to correct the interaction energy may seem at odds with the well-known strong variations with cluster size of the computed chemisorption energy. In recent reviews, Hofmann et al.[21] and Jones et al.[22] discuss approaches to obtain reliable energetics both from calculations under PBC and using cluster models, also including extensions based on embedding techniques. These are very valuable and can be made highly accurate for systems dominated by electrostatics or covalent bonding, but for metallic bonding the cluster-size convergence to zero bandgap is slow and erratic. An example is given by the chemisorption energy of hydrogen on Ni(100) that was found to differ by 9 kcal mol$^{-1}$ when calculated using clusters of 113 and 118 Ni atoms[23]. Efforts to overcome this include the self-consistent embedding by Carter and coworkers[24], but so far only applied to copper. However, copper, as well as silver, has the valence $d$-shell essentially fully occupied and as such does not represent the complexity of transition-metal spin-coupling. Here, we overcome this complexity and present a systematic and general approach that is applicable to both molecular and dissociative adsorption, as well as to challenging open-$d$-shell transition metals, without the need for embedding. In terms of philosophy our approach is equivalent to ONIOM[25] with the reference state being the full periodic DFT calculation.

Similarly to ref. 19 for CO/MgO(001) and to Hu et al.[20] for CO/Cu(111), we combine calculations under periodic boundary conditions (PBC) with higher-level calculations using a cluster model. However, to handle the complexity of the $d$-shell spin-coupling we use a minimum-size cluster and rely on the fact that the local bond strength is quite insensitive to cluster size, as has been shown previously[26–29]. Here, we emphasize the distinction between the bond energy (interaction strength), which is the target of our cluster calculation, and the chemisorption energy, which is the target of the complete approach. We

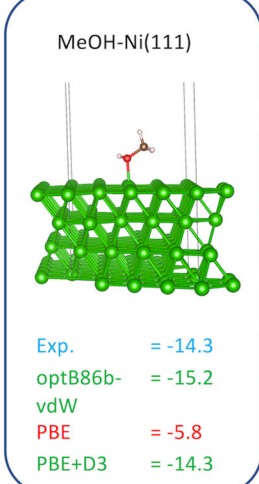
**MeOH-Ni(111)**

Exp.           = −14.3
optB86b-       = −15.2
vdW
PBE            = −5.8
PBE+D3         = −14.3

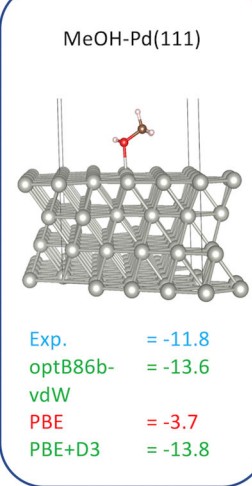
**MeOH-Pd(111)**

Exp.           = −11.8
optB86b-       = −13.6
vdW
PBE            = −3.7
PBE+D3         = −13.8

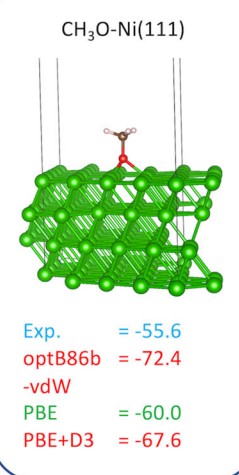
**CH$_3$O-Ni(111)**

Exp.           = −55.6
optB86b        = −72.4
-vdW
PBE            = −60.0
PBE+D3         = −67.6

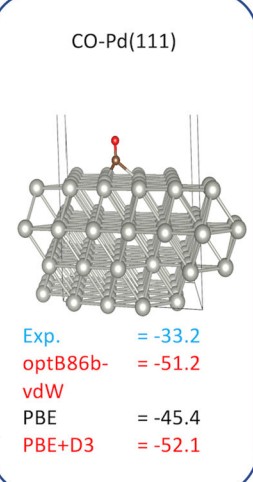
**CO-Pd(111)**

Exp.           = −33.2
optB86b-vdW    = −51.2

PBE            = −45.4
PBE+D3         = −52.1

**Fig. 1 | Functional dependence of computed adsorption energies.** Calculated adsorption energies (kcal mol$^{-1}$) of methanol (MeOH), methoxy (CH$_3$O) and CO on Ni(111) (green slab) and Pd(111) (gray slab) together with the experimental values. Experimental data for methanol and methoxy adsorption energies on Ni(111) were obtained from ref. 72, MeOH on Pd(111) was obtained from refs. 73–75, while the CO adsorption comes from ref. 4.

use the chemical concept of bond-preparation[30] based on earlier studies[26–29] to argue that the local bonding is well-described already by small-size clusters and thus that a higher-level correction can be established. We show it explicitly for CO/Cu(111) in the "Methods" section below, but will first discuss the origin of the variations in chemisorption energy with cluster size.

The cluster is in reality a small molecule with discrete energy levels, which may or may not include a sizeable band gap. This leads to, e.g., the well-known even–odd alternations in reactivity and other properties in cluster physics. These are direct and real physical effects due to the confinement of the electrons in a finite volume of space (particle in a box) and lead to slow and erratic convergence of the computed chemisorption energy with cluster size[23] when computed as the difference between the ground state of the cluster and the chemisorbed system. However, in order for the bond between the surface and adsorbate to form, the electronic structure of both surface and adsorbate must change, which can only be achieved through mixing with excited states. For the extended metal surface, such excitations occur around the Fermi level at zero or small energy cost. For the metal cluster, however, a finite excitation energy may or may not have to be invested to reach the bonding state, and this rehybridization cost reduces the resulting computed chemisorption energy. By calculating this excitation energy for the cluster and adding it to the computed chemisorption energy, surprisingly accurate binding energies can be obtained also from quite small cluster models[26–29]. Thus, the local bond is well defined already for small clusters. However, since the chemisorption energy includes the cost to excite the cluster to the bonding state, and this varies strongly with cluster size and shape, the result is the observed strong variations. Here, we want to emphasize once more that the erratic convergence of the computed chemisorption energies is not an issue to the corrective approach (as shown in "Methods"). By taking the difference of adsorption energies at low and high level of theory, the correction becomes largely independent of the size since the effects related to cluster geometry tend to cancel out.

The application of excited states to compute adsorption energies is also predictive in terms of possible bonding schemes of an adsorbate to a surface. A σ-bond between a hydrocarbon and Pt (in Pt(111)) has been estimated to 53 kcal mol$^{-1}$ by Carter and Koehl[31] and used to distinguish between reaction mechanisms in the decomposition of ethylene on Pt(111) by comparing the energy cost to reach the involved adsorbate excited states in the gas phase with the gain from bond formation. A similar estimate of ~50 kcal mol$^{-1}$ was obtained for unsaturated hydrocarbons on Cu surfaces by Triguero et al.[32] by explicitly taking into account the π→π* excitation energy to reach the bond-prepared di-radical state that can form the two σ-bonds required for a lying-down geometry. For CO and N$_2$, the cost for this excitation is too high to be offset by two σ-bonds, resulting in vertical chemisorption unless additional interactions are available[2,33]. Here, we use this concept to remind the reader why the local chemical bond can be well described by cluster models[26–29], while the chemisorption energy requires special consideration. In addition, we apply it to the cases of chemisorption of benzene and CH on Pt(111).

We will thus use small cluster models, on which higher-level computational techniques can be applied at low cost, to correct the description of the local chemical bond to the surface. The correction is based on the difference in the computed interaction energies in the same geometry and electronic state between the DFT model and the higher-level approach. By taking the difference, the correction becomes largely independent of the size of the cluster and, by retaining the optimized geometry from the periodic DFT calculation, it becomes a correction to the interaction energy in the geometry relevant for the periodic calculation. Combining this with the advantages of the periodic calculation to include effects of band structure, coverage and coadsorbates, as well as localizing transition states, holds the potential to provide accurate and reliable energetics that are independent of the choice of functional. We note, finally, that, since the aim here is only to correct the local bond using the cluster calculations, it is sufficient to ascertain that the same electronic (and therefore spin) states are used both in the higher-level and simpler DFT calculations on the cluster.

Ideally, pure QC techniques, like, e.g., multireference coupled cluster[34–36] or RASPT2[37–39], would be used in the cluster calculations, but to properly account for both dynamical and static correlation effects in a metal cluster of ~10 open-d-shell atoms is still a formidable task with these wave function techniques. This may be alleviated by recent developments in multiconfiguration DFT that hold promise[40–42]. Here, we benchmark a hybrid scheme that combines the advantages of a fast DFT functional for the periodic boundary condition (PBC) calculations and a more accurate hybrid functional applied to a finite-size cluster model. The PBC calculation captures the effects of the band structure and lateral interactions between adsorbates, while the hybrid approach contributes to properly describe the local chemical bond. The energy of the investigated system is thus obtained through an additive scheme[43]. The method can be useful to detect possible errors from currently used approximations and making sure that these are small enough to deliver accurate results or even applied as the method of choice to investigate reactions on transition metal surfaces including accurate transition states. Here, we use PBE+D3 in the PBC calculations and the M06 hybrid functional for the correction of the local bond strength and denote the approach PBE+D3/M06 based on the applied functionals.

## Results

We initially note that benchmarking of computational techniques, here specifically DFT functionals, can have different aims. If the target is to correctly describe dissociative chemisorption of a molecule from gas phase, then the energy difference between the gas phase molecule and the dissociated products at the surface should be computed at the same level and compared directly to the experimentally measured exothermicity of the reaction. Since it is a difference taken between two DFT energies, a good resulting energy may be due to an equally poor description of the system in the gas phase as at the surface. This error cancellation may not be independent of bonding mode, and thus potentially lead to an imbalance when considering further reactions between various dissociated species at the surface, if they are not described equally well throughout the reaction sequence.

An alternative approach, taken here, is to focus on the bond strength in the interaction of the relevant species with the surface, molecularly or dissociatively chemisorbed, which puts the focus more on subsequent reaction steps on the catalytic surface. In order to benchmark relative to experiment, the experimentally measured energy release must then be converted to a gas phase dissociated reference for dissociative cases, which can be done using highly accurate, experimentally determined gas phase dissociation limits. Also in this case a computational gas phase reference is needed, the only difference being that it is now a free atom or radical. When comparing studies benchmarking functionals and computational approaches, we find that this distinction between reference states is essential.

We start by systematically comparing the errors (averaged deviations—MAE and RMSE) produced by the PBE+D3/M06 approach and state-of-the-art methods currently used to calculate adsorption energies on transition metal surfaces with respect to the 38 investigated reactions (Table 1). The DFT energy deviations for the PBE+D3/M06 approach yielded a MAE of 2.4 kcal mol$^{-1}$ and a RMSE of 2.9 kcal mol$^{-1}$. When breaking the dataset into categories: (i) chemisorbed and (ii) physisorbed (Fig. 2 and Tables 1 and 2), we find for chemisorption a MAE of 2.2 kcal mol$^{-1}$ and a RMSE of 2.5 kcal mol$^{-1}$ while for the dispersive interactions MAE is 2.7 kcal mol$^{-1}$ and RMSE is 3.5 kcal mol$^{-1}$. The proposed PBE+D3/M06 approach not only delivers

**Table 1 | MAE and RMSE for chemisorption (Chem.), physisorption (Phys.) and combined (Tot.) using PW91, PBE, RPBE, GAM, MN15-L, BEEF-vdW, SCAN, PBE+D3 and PBE+D3/MO6**

|            | PW91[a] | PBE[b] | RPBE[a] | GAM[b] | MN15-L[b] | BEEF-vdW[a] | SCAN[c] | PBE+D3[d] | PBE+D3/MO6[d] |
|------------|------|------|------|------|------|------|------|------|------|
| MAE-Chem.  | 8.5  | 9.0  | 5.3  | 6.2  | 7.9  | 4.4  | 11.0 | 13.9 | 2.2  |
| MAE-Phys.  | 10.1 | 10.0 | 13.9 | 11.5 | 3.7  | 7.0  | 5.1  | 5.7  | 2.7  |
| MAE-Tot.   | 9.1  | 9.4  | 8.2  | 8.0  | 6.5  | 5.3  | 9.0  | 11.1 | 2.4  |
| RMS-Chem.  | 11.7 | 9.0  | 8.7  | 8.5  | 10.3 | 7.1  | 14.1 | 16.9 | 2.5  |
| RMSE-Phys. | 11.2 | 11.2 | 15.6 | 13.8 | 4.2  | 8.9  | 6.2  | 6.2  | 3.5  |
| RMSE-Tot.  | 11.5 | 12.2 | 11.9 | 10.8 | 8.7  | 7.8  | 11.9 | 14.3 | 2.9  |
| \|MAX\|    | 40.9 | 46.3 | 36.1 | 35.6 | 30.8 | 23.4 | 48.6 | 46.2 | 7.6  |

The maximum absolute deviation (|MAX|) is also shown for each of the functionals. All energies in kcal mol⁻¹.
[a]From ref. 4.
[b]From ref. 9.
[c]From ref. 7.
[d]Present work.

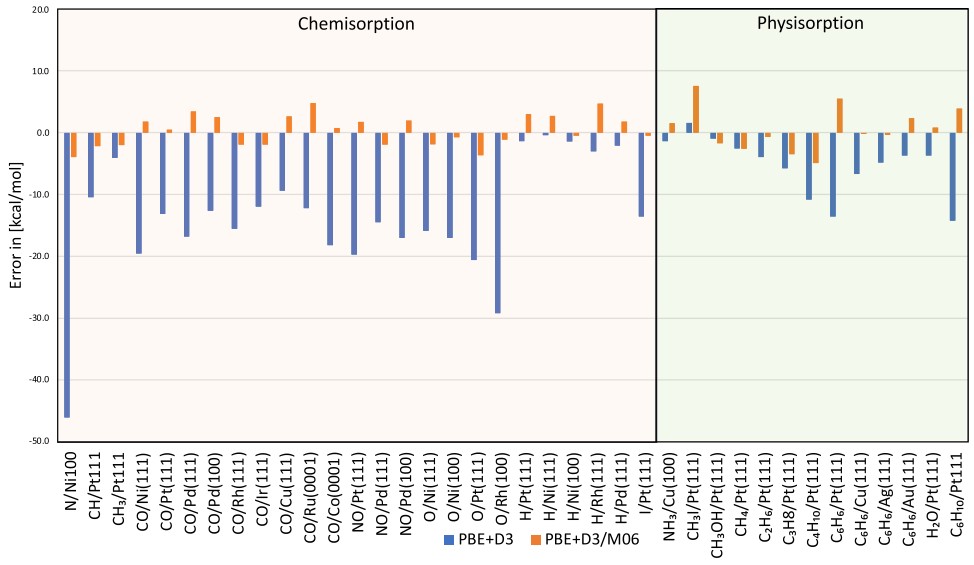

**Fig. 2 | Adsorption energy errors before and after cluster correction.** Error comparison between adsorption energies using PBE+D3 (blue) and the corrected values based on our approach (orange). Systems to the left (N/Ni(100) to I/Pt(111)) are characterized as chemisorption (light purple background fill) while systems on the right (NH₃/Cu(100) onwards) are dominated by van der Waals interactions—physisorption (light green background fill). Source data are provided as Source data file.

generally low deviations as compared to other approaches (Table 1) but also an accurate and balanced description of both kinds of molecule–surface interaction.

Amongst the investigated functionals found in the literature, BEEF-vdW is reported to yield the best average performance with a MAE of 5.3 kcal mol⁻¹ and RMSE of 7.8 kcal mol⁻¹. However, if one separately investigates the performance of BEEF-vdW for chemisorption and physisorption interactions (Table 1), physisorbed systems are persistently poorly described and the good performance is due to the accurate description of chemisorption, hence producing an unbalanced approximation[4,7]. Regarding the chemisorbed adsorbates, the RPBE functional emerges with performance similar to BEEF-vdW, but physisorption is deteriorated—similarly to BEEF-vdW.

Very close in performance to BEEF-vdW is the MN15-L meta-GGA. This functional is from the same group as M06 and was designed to deliver good accuracy for a broad range of applications and moreover tested on a number of properties, such as, e.g., ionization energy, reaction barriers[44]. MAE of MN15-L is 6.5 kcal mol⁻¹—higher than BEEF-vdW. However, MN15-L shows better performance than BEEF-vdW to describe physisorption. The SCAN metaGGA is also an alternative with MAE and RMSE of 9.0 kcal mol⁻¹ and 11.9 kcal mol⁻¹. These functionals—BEEF-vdW and MN15-L—have shown reasonable

deviations in predicted adsorption energies. Yet, the resulting unbalanced errors may become an issue when dealing with complex reactions where both physisorption and chemisorption exist. The PBE+D3/M06, hence, is a viable alternative approach that delivers balanced adsorption energies for chemisorbed and physisorbed adsorbates with a small MAE although it implies extra calculation steps as compared to the standard protocol. Most importantly, the maximum absolute deviation (7.6 kcal mol⁻¹) is by far the smallest in the comparison in Table 1.

Next, we compare the adsorption energies calculated with PBE +D3 with the PBE+D3/M06 results to show the improvements obtained from the finite-size cluster correction (Fig. 2 and Table 2). Large errors for chemisorption energies are observed for PBE+D3. In general, pure PBE performs somewhat better for chemisorption—albeit still overestimating the binding energies[4]. The addition of dispersion corrections, PBE+D3, improves the adsorption energy of physisorbed systems with respect to the pure PBE (see a relatively low error in the range of non-covalent bonds in Fig. 2), but on the other hand, it deteriorates the chemisorption energies with errors that can reach up to 29.3 kcal mol⁻¹ (O adsorbed on Rh(100), for example). The DFT energy deviations yielded a MAE of 11.1 kcal mol⁻¹ for the PBE+D3 approach against a MAE of 2.4 kcal mol⁻¹ for PBE+D3/M06. This

**Table 2 | Difference (theory–experiment) between the reference adsorption energies [kcal mol$^{-1}$] and the values obtained with PW91, PBE, RPBE, GAM, MN15-L, BEEF-vdW, SCAN, PBE+D3 and PBE+D3/M06**

| Reaction | PW91[a] | PBE[b] | RPBE[a] | GAM[b] | MN15-L[b] | BEEF-vdW[a] | SCAN[c] | PBE+D3[d] | PBE+D3/M06[d] |
|---|---|---|---|---|---|---|---|---|---|
| N+Ni(100) → N/Ni(100) | **−40.9** | **−46.3** | **−31.1** | −10.7 | **−30.8** | **−23.4** | **−48.6** | **−46.2** | −3.9 |
| CH+Pt(111) → CH/Pt(111) | 7.1 | 9.2 | 22.2 | 24.9 | 9.4 | 20.2 | −14.2 | −10.5 | −2.2 |
| CH$_3$+Pt→ CH$_3$/Pt(111) | 3.5 | 3.9 | 12.5 | 12.1 | 3.2 | 6.4 | −13.3 | −4.1 | −2.0 |
| CO + Ni(111) → CO/Ni(111) | −12.7 | −14.7 | −3.8 | −22.3 | 12.0 | −5.3 | −13.8 | −19.6 | 1.8 |
| CO + Pt(111) → CO/Pt(111) | −7.6 | −9.1 | −2.9 | −2.5 | −4.1 | −2.2 | −15.1 | −13.1 | 0.5 |
| CO + Pd(111) → CO/Pd(111) | −10.5 | −12.4 | −2.2 | −7.1 | −4.3 | −2.9 | −16.1 | −16.8 | 3.5 |
| CO + Pd(100) → CO/Pd(100) | −5.7 | −6.4 | 0.7 | −1.3 | 0.5 | 1.7 | −10.7 | −12.7 | 2.5 |
| CO + Rh(111) → CO/Rh(111) | −10.3 | −9.8 | −5.0 | −6.5 | −4.8 | −4.8 | −13.1 | −15.6 | −1.9 |
| CO + Ir(111) → CO/Ir(111) | −6.2 | −6.3 | −1.2 | −4.3 | −0.7 | −1.4 | −7.8 | −12.0 | −1.9 |
| CO + Cu(111) → CO/Cu(111) | −4.3 | −3.7 | 0.2 | 2.0 | 9.4 | 0.5 | −7.4 | −9.4 | 2.7 |
| CO + Ru(0001) → CO/Ru(0001) | −5.5 | −5.9 | −0.2 | −0.5 | 1.0 | −0.2 | −6.5 | −12.2 | 4.8 |
| CO + Co(001) → CO/Co(001) | −10.3 | −9.9 | −4.3 | −2.2 | 5.9 | −4.5 | −11.3 | −18.2 | 0.8 |
| NO + Pt(111) → NO/Pt(111) | −13.9 | −12.2 | −6.9 | −4.5 | −13.6 | −7.2 | −15.6 | −19.8 | 1.8 |
| NO + Pd(111) → NO/Pd(111) | −8.4 | −10.4 | −1.4 | −5.3 | −8.9 | −1.0 | −11.2 | −14.5 | −1.9 |
| NO + Pd(100) → NO/Pd(100) | −9.8 | −12.0 | −2.2 | −6.1 | −13.5 | −2.6 | −11.5 | −17.1 | 2.0 |
| O+ Ni(111) → O/Ni(111) | −7.6 | −6.4 | 4.8 | −2.6 | −5.9 | 5.2 | −2.4 | −15.9 | −1.9 |
| O+ Ni(100) → O/Ni(100) | −8.7 | −8.9 | 4.1 | 7.6 | −2.0 | 4.3 | −8.1 | −17.1 | −0.7 |
| O + Pt(111) → O/Pt(111) | −12.3 | −13.0 | −0.3 | 4.2 | 3.5 | −0.6 | −8.7 | −20.7 | −3.7 |
| O + Rh(100) → O/Rh(100) | −19.0 | −19.5 | −6.2 | −3.1 | −6.6 | −6.6 | −14.3 | −29.3 | −1.1 |
| H + Pt(111) → H/Pt(111) | −0.4 | 1.2 | 2.8 | 3.8 | 9.6 | 1.8 | −4.1 | −1.4 | 3.0 |
| H + Ni(111) → H/Ni(111) | 1.4 | 1.7 | 4.6 | −7.6 | 4.6 | 3.2 | −2.8 | −0.4 | 2.7 |
| H + Ni(100) → H/Ni(100) | 0.5 | 1.1 | 4.3 | 4.2 | 14.7 | 2.6 | −4.5 | −1.4 | −0.5 |
| H + Rh(111) → H/Rh(111) | −2.5 | −0.9 | 0.6 | 2.2 | 9.3 | −0.5 | −5.4 | −3.0 | 4.7 |
| H+ Pd(111) → H/Pd(111) | −0.8 | 0.5 | 2.3 | 5.8 | 17.7 | 1.7 | −5.9 | −2.1 | 1.8 |
| I + Pt(111) → I/Pt(111) | −2.5 | −0.6 | 5.5 | 1.6 | −1.8 | 0.3 | −2.8 | −13.6 | −0.5 |
| NH$_3$ + Cu(100) → NH$_3$/Cu(100) | 4.3 | 3.9 | 5.7 | 6.8 | 2.0 | 4.5 | 0.9 | −1.4 | 1.6 |
| CH$_3$I + Pt(111) → CH$_3$I/Pt(111) | 13.9 | 14.7 | 16.3 | 16.8 | 6.8 | 11.5 | 11.1 | 1.6 | **7.6** |
| CH$_3$OH + Pt(111) → CH$_3$OH/Pt(111) | 8.4 | 8.6 | 9.3 | 10.5 | −0.6 | 5.7 | −3.7 | −0.9 | −1.7 |
| CH$_4$ + Pt(111) → CH$_4$/Pt(111) | 2.6 | 2.5 | 3.1 | 1.6 | −2.6 | −0.2 | 1.5 | −2.5 | −2.6 |
| C$_2$H$_6$ + Pt(111) → C$_2$H$_6$/Pt(111) | 5.3 | 5.8 | 5.7 | 3.9 | 3.8 | 1.4 | 3.3 | −3.9 | −0.7 |
| C$_3$H$_8$ + Pt(111) → C$_3$H$_8$/Pt(111) | 7.9 | 8.1 | 8.4 | 6.2 | −0.7 | 2.4 | 4.7 | −5.8 | −3.5 |
| C$_4$H$_{10}$ + Pt(111) → C$_4$H$_{10}$/Pt(111) | 9.3 | 10.2 | 10.0 | 7.5 | −3.8 | 2.4 | 5.3 | −10.8 | −4.9 |
| C$_6$H$_6$ + Pt(111) → C$_6$H$_6$/Pt(111) | 17.0 | 16.0 | 36.1 | **35.6** | −0.7 | 20.4 | −6.2 | −13.6 | 5.5 |
| C$_6$H$_6$ + Cu(111) → C$_6$H$_6$/Cu(111) | 9.3 | 14.6 | 11.0 | 12.8 | −7.6 | −3.8 | 8.9 | −6.7 | −0.2 |
| C$_6$H$_6$ + Ag(111) → C$_6$H$_6$/Ag(111) | 13.1 | 13.9 | 17.0 | 8.2 | −5.7 | 9.6 | 8.4 | −4.8 | −0.4 |
| C$_6$H$_6$ + Au(111) → C$_6$H$_6$/Au(111) | 20.8 | 15.8 | 22.0 | 12.7 | −2.5 | 12.4 | 7.9 | −3.7 | 2.4 |
| H$_2$O + Pt(111) → H$_2$O/Pt(111) | 8.1 | 8.4 | 12.0 | 9.0 | 5.4 | 8.1 | −2.1 | −3.7 | 0.8 |
| C$_6$H$_{10}$+Pt(111) → C$_6$H$_{10}$/Pt(111) | 11.7 | 6.9 | 23.9 | 17.8 | −5.3 | 9.1 | −2.4 | −14.3 | 3.9 |

Water adsorption was treated by us with a 2/3 ML coverage, while for the other cases a coverage of 1/4 ML is used. For C$_6$H$_6$ adsorption on Pt(111), the slab was extended to 5 layers allowing the top 4 layers to relax (see text). MAX values are highlighted in bold.
[a]From ref. 4.
[b]From ref. 9.
[c]From ref. 7.
[d]Present work.

emphasizes the improvements in the adsorption energies when adding the hybrid calculations with the corrective scheme.

RPBE is a functional designed to properly describe adsorption energies of radicals and small molecules like O, CO and NO on transition metal surfaces[45]. This is, hence, a good reference to further evaluate the accuracy of the corrective PBE+D3/M06 scheme in the chemisorption range investigated here. The MAE of PBE+D3/M06 for chemisorbed molecules is 2.2 kcal mol$^{-1}$ compared to 5.3 kcal mol$^{-1}$ obtained with RPBE. For the cases with dispersive interactions, however, the MAE is 13.9 kcal mol$^{-1}$ for RPBE and 2.7 kcal mol$^{-1}$ for PBE +D3/M06. Although RPBE comes with an accurate description for

chemisorption, the deviations when considering dispersive interactions are rather significant (Table 1), while the corrective scheme maintains a similar level of accuracy for both types of interaction.

The largest error (|MAX|) displayed by the PBE+D3/M06 is 7.6 kcal mol$^{-1}$ for the CH$_3$I+Pt(111)→ CH$_3$I/Pt(111) reaction (Table 2). The functional best describing the adsorption of CH$_3$I is the PBE+D3 with an error of only 1.6 kcal mol$^{-1}$, thus, for this case, the corrective scheme does not improve the description. For most of the functionals, |MAX| corresponds to the adsorption of N on Ni(100) (with the exception of GAM that has |MAX| for benzene adsorption on Pt(111)) with values ranging from 23.4 kcal mol$^{-1}$ for BEEF-vdW up to 48.6 kcal mol$^{-1}$ for

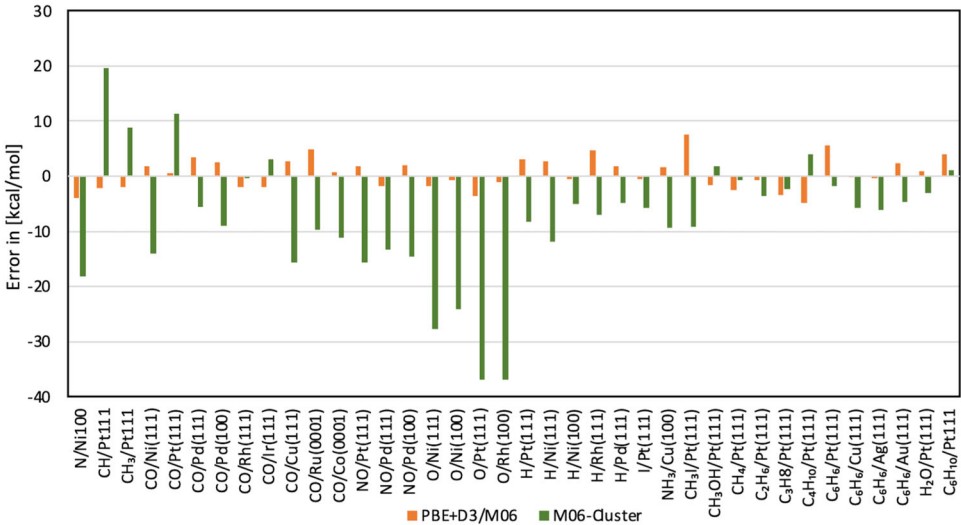

**Fig. 3 | Stability of correction compared to only higher-level calculation on cluster model.** Error comparison (kcal mol⁻¹) between adsorption energies using the PBE+D3/M06 (orange) and the finite-size cluster with M06 (green). Source data are provided as Source data file.

SCAN. In general, the corrective scheme results in lower |MAX| than the other approaches displayed in Table 1, which highlights the predictive power of the approach.

Above, we have discussed the magnitude of the errors when applying pure PBE+D3 under PBC. In Fig. 3 we demonstrate that the significant improvement from the cluster calculations with M06 stems from the improved description of the local bonding and not from a fortunate selection of specific clusters that happened to be in an electronic state corresponding to bond-preparation[30]. The adsorption energy errors with the finite-size clusters and M06 level of theory are, as expected, overall significantly larger than the values obtained with the combination PBE+D3/M06 (see Fig. 3). This is not surprising since no electronic excitations are considered in these calculations—no bond preparation. Yet when such adsorption energies (M06 with small finite-size clusters) are used for the corrective scheme, an accurate description of the adsorption energies is obtained (Table 1). In fact, no excitations are needed for the corrective scheme to be accurate since the goal of the M06 calculations on the finite-size cluster is to improve the description of the local bond that is under/overestimated by the PBE+D3 functional leading to incorrect bond strengths. This correction to the bond strength is obtained in the corrective scheme by the term $E_{HB}^{Ads,Cluster} - E_{PBE+D3}^{Ads,Cluster}$ (Methods Eq. 1).

Since only the local bonding is improved and corrected to the chosen hybrid or QC level of description using the finite-size cluster, the proposed corrective approach is applicable to any functional and holds the potential to make studies independent of the choice of approximate functional. We demonstrate this by applying the correction to PW91, PBE, RPBE and SCAN (Fig. 4 and Supplementary Table 1) instead of PBE+D3. In this case, PBC adsorption energies were taken from the literature (data for PW91, RPBE, and BEEF-vdW were taken from ref. 4 where the dissociative reactions were algebraically modified as in ref. 9, while data for PBE comes from ref. 9) while the cluster corrections were performed as for the case of PBE+D3/M06 keeping the PBE+D3 structures; this is an approximation, but serves to illustrate the point. As seen in Fig. 4, the additive correction results in similar and lower MAEs for the corrected adsorption energies compared to the PBC adsorption energies for all four functionals here tested. For the case of RPBE, the finite-size cluster corrections have reduced the absolute MAE from 8.2 kcal mol⁻¹ (RPBE) to 4.5 kcal mol⁻¹ (RPBE/M06). Breaking the analysis into chemisorption and physisorption, the MAE of chemisorption using RPBE is 5.3 kcal mol⁻¹ while the RPBE/M06 is 3.8 kcal mol⁻¹. On the other

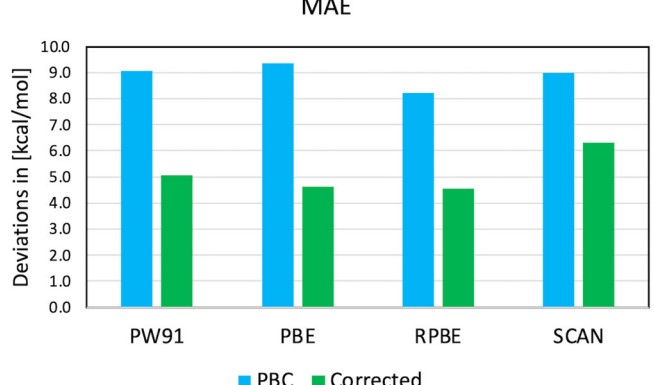

**Fig. 4 | Cluster correction applied to other functionals.** MAE calculated for the PBC-PW91 and corrected PW91/M06, PBC-PBE and corrected PBE/M06, PBC-RPBE and corrected RPBE/M06 and PBC-SCAN and corrected SCAN/M06 where blue corresponds to the MAE of the non-corrected PBC adsorption energies while green stands for the M06 corrected cases. For all cases, the applied correction systematically reduced the obtained MAEs. Source data are provided as Source data file.

hand, MAE's for physisorption with RPBE and RPBE/M06 are 13.9 and 5.7 kcal mol⁻¹, respectively—i.e. a difference of 8.2 kcal mol⁻¹. This is the opposite of the result using PBE+D3 and PBE+D3/M06, for which the difference in MAE's for physisorbed systems is lower than for the chemisorbed ones (PBC-PBE+D3 performs better for physisorption, Fig. 2). The M06 correction thus acts in the range (adsorption vs. chemisorption) where the PBC calculations deliver higher errors and provides a way to get a better balance between physisorption and chemisorption, independent of the chosen functional for the PBC part of the additive scheme.

A practical application challenging the PBE+D3/M06 approach is to properly locate the CO adsorption site on Pt(111)—a well-known puzzle for conventional semi-local functionals where CO is incorrectly located as preferentially being on a fcc hollow site while experiments find the preferred site is on top[46]. The PBE+D3 predicted an adsorption energy of −42.9 kcal mol⁻¹ for CO at the top site of the surface and −46.1 kcal mol⁻¹ at the fcc site. After the correction (PBE+D3/M06) the results are −29.3 kcal mol⁻¹ and −29.2 kcal mol⁻¹ for top and fcc positions, respectively. This confirms that the PBE

+D3/M06 approach properly finds the correct adsorption position of CO on Pt(111), albeit with a very small preference. Hu et al.[20] have earlier shown that applying a similar additive scheme, but with B3LYP as the hybrid functional of choice, predicts the correct CO adsorption site on Cu(111).

The cases of water, CH, and $C_6H_6$ adsorption on Pt(111) merit special attention. Firstly, water adsorbs on the Pt(111) surface forming a hexagonal network with 2/3 coverage[47]. Yet, most of the previously reported calculations discussed here employed a model considering ¼ coverage. Indeed, there are lateral interactions that need to be accounted for (hydrogen bonds) when calculating the adsorption energy of water due to the formed hexagonal structure on the Pt(111) surface. This is reflected by the obtained adsorption energies (computed by us) that, for the case of 2/3 coverage, is −16.8 kcal mol$^{-1}$ while, for the ¼ coverage, is −10.1 kcal mol$^{-1}$ using PBE+D3. After the M06 correction, the calculation employing the 2/3 coverage showed an error of only 0.8 kcal mol$^{-1}$, while the case with ¼ coverage yielded an error of 7.5 kcal mol$^{-1}$. Except for the SCAN functional and PBE+D3, all other approaches yielded errors larger than 5 kcal mol$^{-1}$ (Table 2) for the water adsorption. This is likely due to using an improper structure for the comparison with the experiment.

Benzene can interact with a metal surface as a basically flat, van der Waals-bound molecule or by forming two σ-bonds to the surface in a quinoid (inverted boat) or antiquinoid (boat) structure[32,48]. The electronic structure associated with the benzene quinoid and antiquinoid structures corresponds to excitation from the highest $1e_{1g}$ π-orbital into either of the two components of the unoccupied $e_{2u}$ LUMO π*-orbital, leading to four long and two shorter C-C bonds (quinoid) or two long and four shorter (antiquinoid). These correspond to two possible triplet excited states (on the molecule) with spins localized either on the carbons in para-position (quinoid) or on the other four atoms (antiquinoid) as required to form two σ-bonds to the surface[32]. The excitation energy required to reach this bond-prepared state of the molecule is around 90 kcal mol$^{-1}$ (ref. 32), which may make these structures difficult to find in a standard optimization. Two main changes in the general protocol used in this investigation were thus needed to properly describe the adsorption of $C_6H_6$ on the Pt(111) surface: (i) the benzene molecule was preoptimized on the finite size cluster allowing both cluster and molecule to completely relax during optimization. This resulted in a quinoid benzene structure together with significant reconstruction of the cluster, which indicated that more flexibility is needed also in the PBC calculations. (ii) The optimized structure of the benzene molecule was removed from the Pt cluster and added to a periodic slab model extended to five Pt layers. In the optimization, the bottom layer was fixed and the four top layers relaxed to allow for structural changes as indicated by the cluster model. This is different from other reactions that were treated with a four-layer slab, optimizing only the top two layers. Indeed, we find a reconstruction from three-fold to four-fold hollow of the Pt atoms interacting with the adsorbate and the quinoid structure was retained also in the PBC calculation (see Supplementary Note 1 for the $C_6H_6$/Pt(111) structure). This structure gave an adsorption energy of −53.2 kcal mol$^{-1}$ with the PBC-PBE+D3 approach while the adsorption energy computed in the standard way (four layers Pt with no pre-optimization of the $C_6H_6$ on the finite cluster) resulted in −47.0 kcal mol$^{-1}$. This difference is mainly due to the added flexibility in the slab model with more layers allowed to move–as indicated by the cluster calculation–that allowed the more stable structure to be found. Very interestingly, after the correction, the error coming from the 5 layers slab model for the quinoid state is 4.8 kcal mol$^{-1}$ while for the four-layer slab, resulting in rather undistorted planar benzene, the error even after M06 correction is 11.1 kcal mol$^{-1}$. The combination of PBC and cluster calculations thus allowed the flexibility to find the proper chemisorbed structure for this system. The search for this structure was guided by the expected return on the rehybridization investment,

which could be estimated as (2 σ-bonds = 2*53 kcal/mol)/90 kcal/mol if the necessary two σ-bonds could form[31].

In view of the here performed changes as compared to earlier benchmarks, we recompute the averaged deviations (MAE and RMSE) for RPBE, BEEF-vdW and PBE+D3/M06 by removing the cases: CH/Pt(111), $CH_3$/Pt(111), $H_2O$/Pt(111), $C_6H_6$/Pt(111) and $C_6H_{10}$/Pt(111) (Supplementary Table 2). This allows a direct comparison of the deviations produced by these approaches (RPBE and BEEF-vdW) with the PBE+D3/M06 by considering the same structural models. Generally, the results follow the same trends: (i) RPBE exhibits high MAE and RMSE deviations for physisorption, (ii) BEEF-vdW gives a more balanced description between chemisorption and physisorption than RPBE–but still with better accuracy for chemisorption than physisorption, and iii) neither of these approaches results in better deviations/accuracy than the PBE+D3/M06. This shows that the general analysis regarding the accuracy of the compared approaches/functionals (based on the deviations of Table 1) does not change when these special cases are excluded.

Experimental uncertainty is also a point to consider when evaluating the accuracy of functionals. Table 3 summarizes the found averaged experimental uncertainties from the experimental works provided in ref. 4 (see Supplementary Note 2 for details). For instance, adsorption of CO on Pd(100) has an experimental uncertainty of around 1.1 kcal mol$^{-1}$, hence, functionals providing adsorption energies with an error smaller than 1.1 kcal mol$^{-1}$ can be said to be within experimental uncertainty. The averaged experimental uncertainty (computed from all the found values–Table 3) is 1.6 kcal mol$^{-1}$. This means that there is still room to improve the accuracy of computed adsorption energies since the MAEs of the theoretical approaches vary from 11.1 to 2.4 kcal mol$^{-1}$ (depending on the approach) and are thus overall higher than the experimental averaged uncertainty.

We have also systematically compared the errors (averaged deviations–MAE and RMSE) produced by the PBE+D3/M06 approach and state-of-the-art methods currently used to calculate transition state barriers of dissociative reactions on metal surfaces by selecting five cases from the SBH10 database (results summarized in Table 4 and Supplementary Table 3). The references and data shown for BEEF-vdW were taken from ref. 12, while the results for PBE+D3 and PBE+D3/M06 were computed in this investigation. The DFT energy deviations for the barrier heights yielded MAEs of 2.3 kcal mol$^{-1}$, 10.5 kcal mol$^{-1}$ and 1.6 kcal mol$^{-1}$, for BEEF-vdW, PBE+D3 and PBE+D3/M06, respectively. Sharada et al.[12] have compared the accuracy of meta- and hybrid-functionals like MS2 and HSE06 with BEEF-vdW and confirmed its higher accuracy vs. the counterparts. Very interestingly, they also showed that transition states of these dissociation reactions closely resemble the final states (chemisorbed state), hence, functionals displaying high accuracy to describe the adsorption energies might also deliver accurate activation barriers. The only case presenting accuracy metric even better than BEEF-vdW for the barriers is the PBE+D3/M06 approach. The reason is already mentioned, the PBE+D3/M06 is the approach presenting lowest MAE for chemisorption, hence, also delivering reliable barrier heights. It is worth mentioning that, for dissociative reactions, barrier heights depend linearly on the reaction energy (Brønsted−Evans−Polanyi relations, BEP)[49]. This implies that more exothermic adsorption energies (here more negative) would produce lower activation barriers[50]. Translating these intrinsic relations to our investigation, it is clear from Fig. 2 that the PBE+D3 approach strongly overestimates the chemisorption energies. Therefore, the results presented for the activation barriers for this functional become significantly underestimated (Table 4) with a MAE of 10.5 kcal mol$^{-1}$. Furthermore, as expected, the PBE+D3/M06 shows very high accuracy for the activation barriers, and this can now be assigned to the improved description of the local bond leading to both reliable adsorption energies and activation barriers.

**Table 3 | Coverage, adsorption site, reference energies (experimental values with zero-point energies using PBE subtracted), PBE+D3 adsorption energies, the adsorption energies from the PBE+D3/M06 approach and averaged experimental uncertainties (A.E.U.) (see Supplementary Note 2; values not found in the literature are marked with dash)**

| Reaction | Coverage ML | Site | Ref. [kcal mol⁻¹] | PBE+D3 [kcal mol⁻¹] | PBE+D3/M06 [kcal mol⁻¹] | A.E.U.[kcal mol⁻¹] |
|---|---|---|---|---|---|---|
| N+Ni(100) → N/Ni(100) | 1/4 | fcc | −100.4 | −146.6 | −104.3 | – |
| CH+Pt(111) → CH/Pt(111) | 1/16 | fcc | −163.2 | −173.6 | −165.3 | – |
| CH₃+Pt→ CH₃/Pt(111) | 1/16 | top | −50.9 | −54.9 | −52.9 | ± 4.8 |
| CO + Ni(111) → CO/Ni(111) | 1/4 | fcc | −29.6 | −49.4 | −28.0 | ± 0.7 |
| CO + Pt(111) → CO/Pt(111) | 1/4 | top | −29.6 | −42.9 | −29.3 | ± 3.4 |
| CO + Pd(111) → CO/Pd(111) | 1/4 | fcc | −34.4 | −51.2 | −30.9 | ± 0.7 |
| CO + Pd(100) → CO/Pd(100) | 1/4 | br | −37.5 | −50.2 | −35.1 | ± 1.1 |
| CO + Rh(111) → CO/Rh(111) | 1/4 | top | −33.9 | −49.5 | −35.8 | ± 1.5 |
| CO + Ir(111) → CO/Ir(111) | 1/4 | top | −39.2 | −51.2 | −41.1 | – |
| CO + Cu(111) → CO/Cu(111) | 1/4 | top | −13.6 | −23.0 | −10.9 | ± 0.5 |
| CO + Ru(0001) → CO/Ru(0001) | 1/4 | top | −38.5 | −50.7 | −33.7 | – |
| CO + Co(001) → CO/Co(001) | 1/4 | top | −28.4 | −46.6 | −27.6 | ± 0.2 |
| NO + Pt(111) → NO/Pt(111) | 1/4 | fcc | −28.4 | −48.1 | −26.6 | – |
| NO + Pd(111) → NO/Pd(111) | 1/4 | fcc | −43.5 | −58.1 | −45.5 | ± 1.4 |
| NO + Pd(100) → NO/Pd(100) | 1/4 | hl | −39 | −56.0 | −37.0 | – |
| O + Ni(111) → O/Ni(111) | 1/8 | fcc | −118.4 | −134.2 | −120.2 | – |
| O + Ni(100) → O/Ni(100) | 1/8 | hl | −123.7 | −140.7 | −124.4 | ± 4.8 |
| O + Pt(111) → O/Pt(111) | 1/18 | fcc | −85.3 | −106.0 | −89.0 | ± 1.7 |
| O + Rh(100) → O/Rh(100) | 1/8 | hl | −102.8 | −132.1 | −104.0 | – |
| H + Pt(111) → H/Pt(111) | 1/8 | top | −63.4 | −64.8 | −60.4 | – |
| H + Ni(111) → H/Ni(111) | 1/8 | fcc | −66.7 | −67.1 | −64.0 | – |
| H + Ni(100) → H/Ni(100) | 1/8 | hl | −65.1 | −66.5 | −65.6 | – |
| H + Rh(111) → H/Rh(111) | 1/8 | fcc | −63.4 | −66.4 | −58.7 | – |
| H + Pd(111) → H/Pd(111) | 1/8 | fcc | −65.5 | −67.6 | −63.7 | – |
| I + Pt(111) → I/Pt(111) | 1/8 | fcc | −55.4 | −69.0 | −55.8 | ± 4.8 |
| NH₃ + Cu(100) → NH₃/Cu(100) | 1/4 | top | −14.3 | −15.7 | −12.7 | ± 0.4 |
| CH₃I + Pt(111) → CH₃I/Pt(111) | 1/4 | top | −20.1 | −18.4 | −12.5 | ± 0.5 |
| CH₃OH + Pt(111) → CH₃OH/Pt(111) | 1/4 | top | −13.1 | −14.1 | −14.9 | ± 0.2 |
| CH₄ + Pt(111) → CH₄/Pt(111) | 1/4 | fcc | −3.3 | −6.0 | −6.1 | – |
| C₂H₆ + Pt(111) → C₂H₆/Pt(111) | 1/9 | fcc | −6.5 | −10.4 | −7.2 | – |
| C₃H₈ + Pt(111) → C₃H₈/Pt(111) | 1/9 | top | −9.3 | −15.0 | −12.7 | – |
| C₄H₁₀ + Pt(111) → C₄H₁₀/Pt(111) | 1/9 | top | −11.5 | −20.1 | −14.1 | – |
| C₆H₆ + Pt(111) → C₆H₆/Pt(111) | 1/9 | br | −38.5 | −52.3 | −33.2 | ± 0.6 |
| C₆H₆ + Cu(111) → C₆H₆/Cu(111) | 1/9 | fcc | −15.8 | −22.4 | −15.9 | ± 1.2 |
| C₆H₆ + Ag(111) → C₆H₆/Ag(111) | 1/9 | fcc | −14.6 | −19.4 | −14.9 | – |
| C₆H₆ + Au(111) → C₆H₆/Au(111) | 1/9 | fcc | −16.7 | −20.5 | −14.5 | – |
| H₂O + Pt(111) → H₂O/Pt(111)* | 2/3 | surface | −13.1 | −16.8 | −12.3 | ± 0.4 |
| C₆H₁₀+Pt(111) → C₆H₁₀/Pt(111) | 1/9 | surface | −29.4 | −43.7 | −25.5 | |

Water and C₆H₆ adsorption were treated as discussed in the text.

Two main advantages of using the hybrid functional in a corrective scheme like the PBE+D3/M06 can be highlighted: i) the averaged error found by the PBC-HSE06 screened hybrid functional (see ref. 7) is reported to be higher than the value of BEEF-vdW. Hence, there is no guarantee of more accurate results when using hybrid functionals in the PBC calculation. Adding exact exchange with the PBC-HSE06 certainly improves the description of local bonds, but it also tends to deteriorate the description of states close to the Fermi level – important for the description of the metallic system[51]. That might cause higher MAE as compared to the other approaches. This issue is circumvented with the PBE+D3/M06 approach since the local bond correction is performed using the finite-size cluster. ii) Hybrid functionals are affordable when applied to finite-size clusters but computationally expensive when applied in PBC calculations.

Despite the improvements provided by the M06 corrections, there are still drawbacks with the application of such an additive scheme. The approach needs extra calculations with the chosen functional and M06 on the finite clusters. That makes it less computationally efficient than PBC-BEEF-vdW or PBC-RPBE. On the positive side, the additive scheme proposed here is very flexible. Computational power is constantly increasing and that should enable, in the future, the application of higher-level wave function based methods like, e.g., CCSD(T) as the corrective approach instead of the hybrid M06, which still relies on an approximate functional.

## Discussion

Here, we propose an approach to compute accurate and reliable energetics for surface-adsorbate interactions and transition states by combining PBC calculations to capture band structure and coverage

**Table 4 | Experimental and calculated reaction barriers (kcal mol⁻¹) and associated errors**

| Reaction | Barriers | | | | Error | | |
|---|---|---|---|---|---|---|---|
| | Exp. Ref. | BEEF-vdW (SBH10) | PBE+D3 | PBE+D3/M06 | BEEF-vdW (SBH10) | PBE+D3 | PBE+D3/M06 |
| H₂ + Cu(111) → H₂/Cu(111) | 14.53 | 16.37 | 3.5 | 15.1 | −1.84 | 11.0 | 0.5 |
| H₂ + Cu(100) →H₂/Cu(100) | 17.07 | 16.6 | 6.9 | 17.2 | 0.47 | 10.2 | 0.2 |
| H₂ + Pt(111) → H₂/Pt(111) | 0.00 | 2.77 | 0.0 | 0.0 | −2.77 | 0.0 | 0.0 |
| CH₄ + Ni(100) →CH₄/ Ni(100) | 17.53 | 20.75 | 5.8 | 20.5 | −3.22 | 11.8 | 3.0 |
| CH₄ + Ni(111) →CH₄/ Ni(111) | 23.29 | 26.29 | 9.4 | 25.1 | −3.00 | 13.9 | 1.8 |
| MAE | – | – | – | – | 2.26 | 9.4 | 1.1 |
| RMSE | – | – | – | – | 2.48 | 10.5 | 1.6 |

References and values for BEEF-vdW are from ref. 12, MAE and RMSE using BEEF-vdW SBH10 reference, and this work PBE+D3 (PBC) and PBE+D3/M06.

effects with cluster calculations using a higher-level approximation to improve the description of the local chemical bond.

We thus investigate and benchmark a hybrid approach named PBE+D3/M06 against 38 experimental reliable adsorption energies. The approach makes use of periodic boundary condition calculations using the PBE+D3 approach together with finite-size cluster calculations performed with the hybrid functional M06 and PBE+D3. In this way, band structure and coverage effects are captured by the PBC calculations and adsorption energies are locally corrected for the errors inherent from the semi-local GGA by using a hybrid approximation. The results showed a MAE of 2.4 kcal mol⁻¹ and an RMSE of 2.9 kcal mol⁻¹ with the only approach that performs similarly being BEEF-vdW with MAE and RMSE of, respectively, 5.3 and 7.8 kcal mol⁻¹. Moreover, PBE+D3/M06 provides similar accuracy in treating both chemisorbed and physisorbed systems−delivering a balance between adsorbates with different dominating surface interactions. The good performance of the PBE+D3/M06 in comparison to currently used approaches, like BEEF-vdW or RPBE, thus potentially places it as an alternative for applications to heterogeneous catalysis and electrocatalysis. As for activation barriers of dissociative reactions the PBE+D3/M06 had MAE and RMSE of 1.1 and 1.6 kcal mol⁻¹, respectively, again better than BEEF-vdW at 2.3 and 2.5 kcal mol⁻¹. This is due to the high accuracy revealed by the approach to describe chemisorbed systems that, due to the BEP relations, leads to accurate activation barriers. The good performance of the PBE+D3/M06 in comparison to currently used approaches, like BEEF-vdW or RPBE for adsorption energies and also barrier heights, thus potentially places it as an alternative for applications to heterogeneous catalysis and electrocatalysis.

The additive approach behind the PBE+D3/M06 is furthermore applicable to correct predicted interaction energies (adsorption energies and activation barriers) from any functional to the desired higher-level accuracy without the drawback of performing the higher-level calculation on the extended PBC system. In practice, this holds the potential to make results largely independent of the choice of functional as long as a reliable higher-level approach is available. What is required is extracting a cluster with the adsorbate from the PBC calculation so that the local geometrical structure is identical between PBC and cluster. The constraint on the cluster calculation is that the electronic structure (e.g., spin state and spin-coupling within the cluster) should be the same for both the lower- and higher-level calculations. With further developments of computer power and quantum chemical wave function techniques, it can be expected that the required cluster calculations, also for transition metals, can be made independent of approximate functionals and driven to "arbitrary accuracy." Furthermore, since the correction works on the local interaction between metal and adsorbate, it is equally applicable to verify or correct computed barriers in a reaction sequence, where high accuracy is particularly essential for reliable microkinetic modeling of rates and turn-over frequencies.

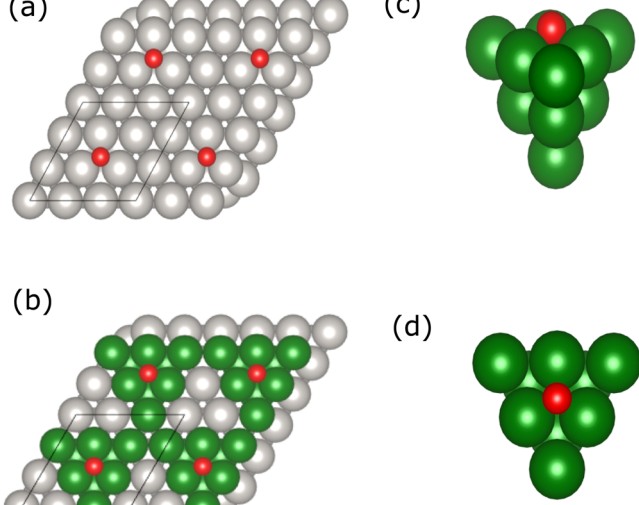

**Fig. 5 | Extraction of cluster from periodic calculation.** O adsorbed on Pt(111) (**a**). Selection of the cluster (green) used in the corrective scheme (**b**). Cluster with O/Pt atoms (side (**c**) and top (**d**) views) obtained from the periodic structure.

## Methods

Our hybrid scheme works as follows

(i)   Perform the adsorption energy calculations with PBC using a GGA functional (here PBE+D3). The adsorption energy ($E_{PBE+D3}^{Ads,PBC}$) is defined as $E^{Ads} = E\left(\frac{A}{M}\right) - E(A) - (M)$, where A and M refer to the adsorbate and metal surface, respectively.

(ii)  Perform adsorption energy calculations with a finite-size cluster with the hybrid functional (HB) and the same GGA functional (PBE+D3) as used in the PBC calculation ($E_{HB}^{Ads,Cluster}$, $E_{PBE+D3}^{Ads,Cluster}$). Here, the pure metal clusters are obtained from the pure PBC metal surfaces. The adsorbate/clusters are obtained from the adsorbate/surface PBC-optimized structures. For both clusters and adsorbate/clusters the PBC-optimized structures are retained while the gas-phase molecules are structurally optimized.

(iii) Employ Eq. (1) to assess the corrected $E_{ads}$:

$$E_{ads} = E_{PBE+D3}^{Ads,PBC} + E_{HB}^{Ads,Cluster} - E_{PBE+D3}^{Ads,Cluster} \qquad (1)$$

In such a way, the band structure and effects of coverage are captured by the PBC calculations while the local chemical bond is improved with the hybrid functional of choice. Figure 5 displays how the finite-size clusters are obtained from the periodic structure for the case of O adsorbed on Pt(111). The clusters for all the other metals (Au, Ag, Ni, Pt, Pd, Rh, Cu, Ru, Ir and Co) are constructed following the example in Fig. 5.

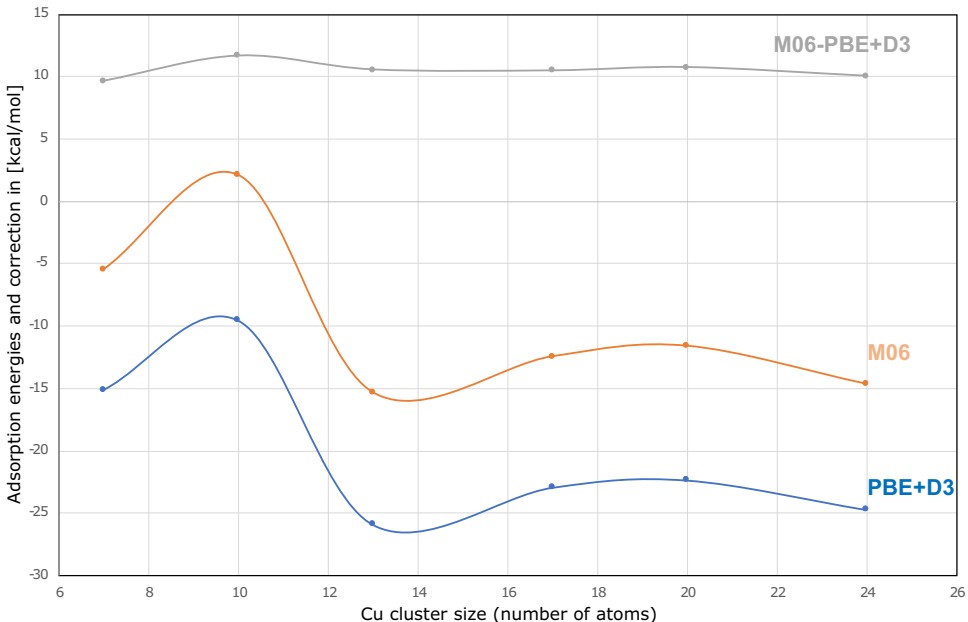

**Fig. 6 | Cluster correction depends weakly on cluster size.** Adsorption energy of CO on Cu clusters with increasing number of atoms using M06 (orange) and PBE +D3 (blue). The employed correction to the PBC adsorption energy, E(M06)-E(PBE +D3), is shown in gray. The variation of the adsorption energy vs. the number of atoms in the Cu clusters with M06 and PBE+D3 is similar, hence, the corrective term becomes almost constant. Lines are guides to the eye. Source data are provided as Source data file.

Two points were considered when building the clusters (Supplementary Fig. 1). Firstly, adsorbates must be coordinated (first shell) similarly to the extended surface calculation. Secondly, the calculation of the term ($E_{HB}^{Ads,Cluster} - E_{PBE+D3}^{Ads,Cluster}$) has to be performed with clusters in the same spin state and coupling for the PBE+D3 and HB calculations. For open $d$-shell metals like Ni it then becomes necessary to limit the cluster size in order to avoid spurious effects of inequivalent electronic structures that become difficult to control for larger cluster sizes. Thus, the focus has been to use clusters that are electronically similar at the two levels of calculation rather than using the same cluster size and shape to represent all structurally similar surfaces, e.g., Pd(111) and Ni(111), which are rather different in terms of electronic structure. This is based on the fact that, since the correction applies to the local bond and is obtained as the difference between two calculations, it becomes rather insensitive to the cluster size and shape. To demonstrate this, we consider CO adsorption on the top site of Cu(111); using copper eliminates issues around the spin-coupling and allows easy extension to large clusters.

We have used clusters with 7, 10, 13, 17, 20 and 24 Cu atoms to represent Cu(111) (Fig. 6). Between the 7- and 24-atoms clusters, we find a difference of 0.5 kcal mol$^{-1}$ in the correction, in spite of a chemisorption energy difference of 9.2 kcal mol$^{-1}$. The largest difference in chemisorption energy between any of the clusters is 16.1 kcal mol$^{-1,}$ with a difference in the correction of 2.1 kcal mol$^{-1}$. There is thus a slight dependence on the cluster size, but the variation is within the error bars of typical experiments (Table 3). We can thus select clusters that are small enough to reliably identify the proper electronic states with reasonable computational effort ensuring robustness to the approach, but still without losing accuracy.

**The PBC calculations**
PBC calculations were performed using the projected augmented wave method as implemented in the Vienna ab initio Simulation Package (VASP)[52,53]. For the optimizations, the wave functions were expanded using plane-waves with a cutoff energy of 500 eV for the valence electrons and a (4×4×1) $k$-point mesh. At the optimized geometry, the self-consistent energies were recomputed with a cutoff energy of 700 eV and an (8×8×1) $k$-point mesh which has been shown to give converged results in earlier work[4,9]. Partial occupations were obtained using the Methfessel–Paxton scheme of order 2 with a smearing of 0.2 eV. A four-layered slab was used to model the metal surfaces where the two topmost layers were allowed to optimize while the two bottom layers were fixed at the optimized bulk structure; this is the same approach as in previous benchmark studies. A vacuum of 20 Å was used to avoid interaction between the periodic images. The effects of dispersion in the interaction between the metal surface and the adsorbates were accounted for with the D3 semiempirical approach of Grimme with the Becke-Johnson (BJ) damping function[11,54]. An improved (D3$^{Surf}$) approach that reduces the polarizability for highly-coordinated atoms has recently been proposed, but so far with parameters only for Cu and Ag[55]. The exchange and correlation term of the Kohn–Sham Hamiltonian is described with the PBE functional[18]. Adsorption sites and coverages were taken from the work of Wellendorff et al.[4]. Exceptions are water adsorbed on Pt(111) for which we used a 2/3 ML coverage forming a hydrogen-down structure[47], CH adsorption on Pt(111) and $CH_3$ adsorption on Pt(111) for which we used 1/16 ML coverage to better mimic the experiment. Moreover, the adsorption of $C_6H_6$ on Pt(111) and $C_6H_{10}$/Pt(111) were treated using a slab of 5 layers where the top 4 layers were allowed to optimize; the reason for this is discussed in the Results section. Dipole correction was included in the direction of the surface normal. Bulk optimizations were performed with a $k$-point mesh of 15x15x15 for the fcc structures and 15×15×13 for the hcp structures. The obtained lattice parameters are summarized in Supplementary Table 4.

**The hybrid functional choice**
The choice of the hybrid functional is vital to develop a scheme that shows a balance between chemisorption and physisorption. Two systems were investigated: (1) CO adsorption on Pd(111) and (2) $CH_3OH$ adsorption on Pd(111). $E_{ads}$ were computed in such a way that $E_{HB}^{Ads,Cluster}$ was treated with different hybrid functionals (see Supplementary Figure 2). From these calculations, the M06 hybrid functional emerged with the highest accuracy for both reactions[56]. This functional is part of a set of functionals known as Minnesota functionals.

Specifically, the M06 functional was developed to treat highly correlated systems and also deliver a reasonable description of non-covalent interactions—dispersion interactions are partially accounted for through its parametrization. Hence, it is expected that this meta-hybrid would provide a balanced and accurate description of covalent and non-covalent interactions. The term $E_{HB}^{Ads,Cluster}$ in Eq. (1) will thus be written as $E_{M06}^{Ads,Cluster}$ and we refer to our approximation as PBE+D3/M06 in reference to the functional used in the PBC calculations and the hybrid used in the finite-cluster calculations.

### Transition states and barriers

Initial transition state (TS) structures, computed using the BEEF-vdW functional, were taken from the SBH10 database[12]. These TS structures were recomputed using PBE+D3 with the optimized lattice parameters used in this work and employing the Dimer method[57]. Barriers were calculated as the energy difference between the transition states and the gas phase molecules plus the clean metal surfaces, as the corresponding reference. All other parameters were the same as for the chemisorption energy calculations.

### The cluster calculations

Cluster calculations were performed using the ORCA package[58] employing the def2-TZVP basis set[59] for both the PBE+D3 and M06 approaches. To speed up the calculations, the Resolution of Identity (RI) in conjunction with def2/J auxiliary basis sets and the Chain of Spheres (COS) approximation were used[60–62]. Effective core potentials of Stuttgart[63] (def2-ecp) type were employed for atoms heavier than Kr—for other cases (Co, Ni and Cu) all-electron calculations were performed. The effects of basis set superposition errors (BSSE) on $E_{ads}$ were evaluated, but found to be similar for PBE+D3 and M06 for the present set of adsorbates and thus cancel out when taking the difference (Supplementary Table 5). We note that using different basis sets in the PBC (plane waves) and cluster (atom-centered Gaussian basis) calculations is a fully consistent procedure since Eq. (1) only contains adsorption energies computed as an energy difference for each system separately. However, naturally, the cluster calculations have to be performed using the same basis set for both the lower- (PBE+D3) and higher-level (M06) approaches in order to be comparable.

We comment briefly on using the same geometry for the PBE+D3 and M06 calculations on the cluster. This is intentional since the M06 is used to obtain an improved estimate of the interaction at the geometry of the PBC calculation. Reoptimizing the adsorbate on the small cluster, even with the cluster atoms fixed, may introduce uncertainties in terms of, e.g., edge effects and lose the connection with the optimized structure in the PBC calculation, which typically includes effects of coverage and coadsorbates.

To select the correct spin multiplicity for the bare cluster, several calculations with possible electronic states were carried out in the PBE+D3 approach to find the multiplicity giving the lowest electronic energy. Subsequently, the clusters with adsorbate were allowed to vary their spin state by small amounts to find the lowest energy state; for the correction it is essential that the higher-level calculation also uses these same spin-states. For instance, spin multiplicity variation of the pure Pt(111) cluster (Fig. 5) showed a multiplicity of 9 as the ground state energy. Therefore, multiplicities of 7, 9 and 11 were tested for the cluster Pt(111)+O (Fig. 5) and the one with lower energy was selected for the adsorption energy calculation—in this specific case, 7 is the lower-energy possibility. This is different from the bond-preparation scheme of refs. 26–29 where the objective was to determine which excited state of the cluster is involved in the bonding. Here the goal is to find a well-defined electronic state that is maximally similar between the PBE+D3 and hybrid cluster calculation (as discussed above). The M06 hybrid functional calculations were thus performed using the same magnetic states as the

PBE+D3 (same multiplicities for both functionals for the pure cluster and cluster+adsorbate). Supplementary Fig. 1 displays the selected clusters together with the adsorbates. In general, the gas phase molecules were treated by using their ground state electronic structure. CH, however, is an exception where the gas-phase ground state is a doublet that cannot form the required three bonds to the surface. The bond-prepared gas-phase reference state thus corresponds to the lowest quartet state, which was used as reference instead of the doublet.

Finally, to emphasize the connection to the surface from which each cluster is derived, we will in the following denote each cluster as the original surface, e.g., Pt(111) cluster rather than $Pt_{10}$; the individual clusters are shown in Supplementary Fig. 1 with coordinates given in Supplementary Note 4.

### Additive scheme with other functionals

We also benchmarked the additive scheme for other functionals in the PBC part. In this case, Eq. (1) becomes: $E_{ads} = E_{Functional}^{Ads,PBC} + E_{M06}^{Ads,Cluster} - E_{Functional}^{Ads,Cluster}$. For the first term, $E_{Functional}^{Ads,PBC}$, the adsorption energies were taken from the literature where the functionals PW91, PBE, RPBE and SCAN were employed. The third term, $E_{Functional}^{Ads,Cluster}$, was computed as described in the previous subsection using the functionals PW91, PBE, RPBE and SCAN[18,45,64,65] and assuming the same multiplicity as in the PBE+D3 calculations. The clusters created for the PBE+D3/M06 approach were used also in these cases.

### The experimental database

The reactions considered were taken from the CE39 data set[4] (39 experimentally determined reactions), as compiled in Table 3. For the cases of dissociative adsorption, the reported energies contain the dissociation of the adsorbate, as well as the adsorption of the radical. Here, adsorption energies of dissociative reactions were algebraically corrected to correspond to the radical adsorption process. For instance, the adsorption of $O_2$ on Pt(111) is read as $O_2+Pt(111)\rightarrow 2O/Pt(111)$ in the work of Wellendorff et al.[4] while here it is treated as $O+Pt(111)\rightarrow O/Pt(111)$. The transformation is performed as the energy of the reaction $O_2+Pt(111)\rightarrow 2O/Pt(111)$ minus that of dissociating the $O_2$ molecule, $O_2\rightarrow O+O$, and dividing by two. Experimental dissociation energies for $O_2\rightarrow 2O$, $H_2\rightarrow 2H$, $I_2\rightarrow 2I$ and $NO\rightarrow N+O$ were employed to convert the reported energies of Wellendorff et al.[4]. The values of the dissociation energies are 120.8, 109.5, 35.9 and 152.7 kcal mol⁻¹, respectively[66–68]. This approach was also employed in ref. 9. The mean absolute error (MAE) and root mean squared error (RMSE) were used to indicate the "goodness" of the employed approach and for comparison with other methods (error, here, is the computed adsorption energy minus the experimental reference). In addition, we report the maximum absolute deviation (MAX) as a measure of predictive power.

The experimental reaction enthalpies were also converted to reaction energies by correcting the zero point energies from PBE calculations and thermal contributions[4]. Hence, we used these vibrationless adsorption energies for the comparison—this has also been the strategy used in the recent literature[4,7–9]. The experimental adsorption energies including these corrections will be denoted reference energies in the following. The employed reference value of the reaction $C_6H_6$/Pt(111) is slightly different from previous works since we used the more recent reaction energy value of −38.4 kcal mol⁻¹ from ref. 69. For the adsorption of CH and $CH_3$ on Pt(111), experimental values were used as the reference where the reported values were corrected by zero point energies from PBE calculations and thermal contributions[70,71]. The reaction $H_2O+1/3O/Pt(111)\rightarrow 2/3(H_2O\cdots OH)/Pt(111)$ was not included here since the minimum size cluster to describe this system exceeds what was deemed feasible with the present approach. Therefore, instead of using all 39 reactions from the CE39 data set[4], we used here 38 cases to benchmark the proposed additive scheme.

## Experimental activation barriers

Five reactions were selected from the SBH10 database[12]: $H_2$ + Cu(111) → $H_2$/Cu(111); $H_2$ + Cu(100) → $H_2$/Cu(100); $H_2$ + Pt(111) → $H_2$/Pt(111); $CH_4$ + Ni(100) → $CH_4$/ Ni(100); $CH_4$ + Ni(111) → $CH_4$/ Ni(111). These reactions were selected as sample of cases where: the metallic cluster has better separated multiplicity states (two Cu surfaces); the cluster has multiple close-lying multiplicity states (two Ni surfaces); and the chemical process is barrierless (Pt surface), but the theoretical reference of the database using the BEEF-vdW functional has a positive barrier. Differently from the previous case, where adsorption energies of dissociative reactions were algebraically corrected to correspond to the radical adsorption process, here, for transition states, no manipulation of the dissociative reactions was performed (we followed exactly as in the SBH10 database, hence, allowing a direct comparison regarding the performance). Moreover, in line with the SBH10 work, all the barriers were corrected by the same ZPE values as included in the reference SBH10 database.

## Data availability

Supplementary information is available in the online version including additional details on the choice of the M06 hybrid functional, the structure of the used clusters, tables containing extra information about the clusters and adsorption energies obtained with all functionals used here. Source data are provided with this paper.

## Code availability

The VASP code is licensed software available from https://www.vasp.at/ while the ORCA code is free to download after registering at https://orcaforum.kofo.mpg.de/app.php/portal.

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

## Acknowledgements
We are grateful for support from the Swedish Foundation for Strategic Research (SSF) through grant number EM16-0010. The Swedish National Infrastructure for Computing (SNIC) provided computational resources at the PDC and NSC centers, which are partially funded by the Swedish Research Council through grant agreement no. 2016-07213.

## Author contributions
L.G.M.P. designed and supervised the study. R.B.A. and G.L.S.R. performed the calculations with input from E.C.d.S. R.B.A. wrote the first draft of the paper. All authors discussed and contributed to the analysis of data and to the final manuscript.

## Funding

## Competing interests
The authors declare no competing interests.
