## [Peer Review File · Nature Communications]

Adsorption Energies on Transition Metal Surfaces: Towards an Accurate and Balanced DescriptionReviewers' comments:

Reviewer #1 (Remarks to the Author):

Araujo et al. present an approach to improve the accuracy of DFT calculations of adsorption energies on transition metal surfaces. They combine periodic calculations with GGA functionals with cluster calculations and hybrid functionals. The results are compared with the 39 available experiments (Wellendorf 2015 and Sharada 2019) showing an excellent agreement: MAE 23 kcal/mol (0.10 eV). The set includes problematic molecules such as CO, together with physisorbed ones such as C1-C4 hydrocarbons, although other families of reactions were omitted. The authors provide enough information to allow reproducibility. Overall, I have found the Manuscript interesting, yet the content needs to be improved to be fully convincing:

The way the authors chose the cluster size raises several questions. Figure S2: 7 atoms for H/Ni100 (a typo duplicates Ni111), 8 for CO/Pd100, 9 for H/Ni111, 10 for H/Pt111...

* First, it was not clear why the cluster size differs for H on Ni111 and Pd111, being both of the same group. I would have expected a unique surface for all fcc(111), another for fcc(100), and yet another for hcp(0001) surfaces.

* Second, and I would like to emphasize this point: The clusters seem exceedingly small. Metal nanoparticles have wild variations in their electronic structure until they reach diameters of around 15Å or 50+ atoms. Some properties may converge with smaller particles. In Reference 38 (10.1103/PhysRevLett.98.176103, see Figure 2 and the text) an early convergence of 16 atoms is reported. Yet, in the present work much smaller particles having less than half atoms were used.

Regarding the set.

* The authors mention databases of 40+ well-defined experimental adsorption energies (Wellendorf 2015 and Sharada 2019), yet only 34 of them were investigated. Why? One wonders why important reactions, such as the dissociative adsorption of CH₃I and CH₂I₂ were omitted (reactions 24, 25, 27 in Wellendorf 2015). The authors seem to have included only the non-dissociative CH₃I adsorptions from Sharada 2019. Including all reactions in the dataset would have shown that the present approach also works for dissociative adsorptions of hydrocarbons, going beyond that of simple diatomic molecules H₂ or O₂.

Regarding the novelty:

* Combining periodic GGA with high-level cluster calculations does not seem fully novel when compared to that of Ref 38. The authors may highlight the novelty.

Reviewer #2 (Remarks to the Author):

The paper reports adsorption energies for a large range of systems calculated with a cluster-corrected approach that is equivalent to the long-standing additive ONIOM scheme (J. Phys. Chem. 1996, 100, 50, 19357–19363) and previously employed by the groups of Sauer, Reuter (Refs. 37, 38), and others not cited such as Carter (e.g. J. Phys. Chem. C 2008, 112, 12, 4649–4657).

As such, the approach itself is not novel. The paper presents novel results, namely adsorption energies for a comprehensive range of reactions and reports excellent average errors across the dataset, which comprises equilibrium structures of small chemisorbed and physisorbed adsorbates on various metal surfaces. The main claim is that this approach can predict highly accurate interaction energies by correcting any functional with cluster calculations performed with any desired high-level functional to achieve "arbitrary accuracy". It is further claimed that the correction is local and should therefore also apply to barriers in reaction sequences.

Unfortunately, I do not find the work convincing. The conclusions do not go significantly beyond the findings of previous works and the manuscript does not provide sufficient information to corroborate the central claims.

1. The method stands and falls with the claim that the correction is local. However, this is not explicitly shown. The authors do not describe the rationale for the shape and size of the clusters that are cut from the surfaces and they do not provide quantitative evidence for the swift convergence of $E^{\text{ads}}_{\text{HB}} - E^{\text{ads}}_{\text{PBE+D3}}$ as a function of cluster size. Note that total energy does not converge smoothly and that previous works showed some locality only for differences of E_{xc} . The choice of cluster size therefore appears arbitrary. In general, no systematic convergence data is provided how for example the different type of basis and size of basis set between the local atomic orbital and periodic plane wave code affect the results.

2. The argument on BSSE is not convincing as higher-level methods typically require larger basis sets with ensuing higher BSSE. If it is negligible then this should be shown. If not, then this and other numerical considerations clearly limit the ability to achieve “arbitrary accuracy” (or rather precision).

3. The error analysis is limited to MAEs and RMSE, but tables in the manuscript report MAX errors. It would be interesting to provide a deeper analysis what structures are associated with these maximum errors and if these structures are the same that provide large errors for conventional functionals. Is the structure of the error distribution of e.g. PBE preserved after correction or is the remaining error of the scheme random/statistical?

4. For anything else than late and noble transition metal clusters, the question of the spin and electronic state of the cluster will become increasingly difficult with a high level of manual intervention. It is not clear from the manuscript how this will be treated for more challenging systems and for nonequilibrium structures (transition states etc). To some extent, the discussion on page 13 around benzene already shows the limitations of this approach.

5. I was not convinced by the qualitative argument around capitalistic chemistry. This relates to penalties due to electronic and geometrical preparation to adsorption, which is not even analysed in this manuscript but only relates to discussions in other works. The argument is that GGAs get the local chemistry, preparation penalty etc. wrong and that those will be the same on cluster and PBC and therefore can be corrected by subtraction? This should be explored in more detail.

6. The authors raise the context of computing accurate barriers, but I am not convinced that this approach will provide as accurate barriers as it does adsorption energies or that it will provide smooth and sensible potential energy landscapes. This should be systematically studied. Transition states (non-equilibrium states in general) exhibit more non-local chemistry than groundstates, so it would be important to know if the same arguments hold.

Reviewer #3 (Remarks to the Author):

I am reviewing the manuscript of Araujo et al for publication in Nature Communication. The work aims at providing a computational recipe for an accurate prediction of adsorption energies on the surface of transition metals, applying a concept they term “capitalistic chemistry”. The target of the work, obtaining highly accurate adsorption energies throughout chemical space, independent of the type of the bonding, is highly important and an important goal for various communities, including surface science, organic electronics, catalysis, and many more. Unfortunately, for the reasons discussed below, I am unable to recommend the publication of this work in the present form.

The analogy with economics and business decisions seems intriguing and appealing at first. However, throughout the work it becomes clear that this frame is just a re-wording of well-known chemical concepts, unnecessarily introducing jargon. The analogy also seems to be incorrectly used

sometimes. For example, the authors define the cost to re-hybridize as energetic investment (with “energy” as unit), and the energetic gain through the formation of the new bond as “return on investment” (which would also have energy as unit). However, while there are various possible definitions on ROIs, it is customary to define it as the ratio between net income and investment (this being unitless). Also, it is usually employed to decide whether a certain investment should be undertaken (when given multiple opportunities), which is where the analogy, again, falls short. While this could be solved by using more appropriate terms, I feel that mixing the jargons of chemistry and economics does not provide any additional insight if it doesn’t introduce a new, beneficial concept. It therefore should be removed completely.

On more scientific grounds, the authors suggest a hybrid scheme of low-level band-structure and high-level cluster calculations. Such QM/QM schemes that add and subtract energies at different levels of theory are a common concept for interface calculations (see, e.g. <https://pubs.rsc.org/en/content/articlelanding/2013/CP/c3cp52321g#!divAbstract>). I am afraid that I fail to see the innovative character of the proposed scheme and why it would be published in Nature Communications. Even if there is an innovative aspect that I am missing, I feel that a comprehensive discussion of earlier works with such hybrid and/or other embedding schemes is missing. Without the innovative character of a hybrid scheme, the present work is mostly a benchmark paper for the performance of various computational schemes for the adsorption of small molecules on transition metals. Such a work is clearly important. Unfortunately, there are also some more technical issues. In order to provide “accurate and balanced” adsorption energies with the given hybrid schemes (and in order to transfer the findings to other systems), it is of paramount importance that the individual terms of eq. 1 are numerically well converged with respect to the computational settings and, therefore, robust.

The authors have mostly used 4-layer slabs throughout the work. I doubt that, even for small molecules, this yields energies that are converged to less than 0.1kcal/mol. Likewise, also a 4x4 k-point grid would be reasonable for a standard investigation, but I am not convinced that it is converged to the 0.1kcal/mol accuracy required to make the claims in that paper. In fact, already the employed broadening scheme in the PBC calculations, 2nd-order MP, may be at the verge of having a too large broadening parameter (0.2 eV) to allow for a sufficiently faithful backextrapolation to zero-broadening (see also <https://doi.org/10.1039/DOCP06605B>) with that accuracy.

Furthermore, I am concerned about the author’s statement that the BSSE between PBE+D3 and HB cancels out. There is no fundamental reason why it should, and in fact, it is well known that more correlated methods show a larger BSSE than semilocal functionals. I would expect that the authors find this cancellation here only because the molecules they investigated were very small; I very much doubt that it will hold for larger systems (where the BSSE becomes larger in absolute terms). Along the same lines, I am concerned about the use of different basis sets for PBC and cluster calculations. I understand that, to some extent, this happens out of necessity. Not many codes can deal with both clusters and PBCs with the same basis set, but some can. However, the different limitations of the two basis sets (plane wave with 500 eV cutoff in a given unit cell versus LCAO triple-zeta) is likely to introduce a certain systematic bias in the benchmark. I would suspect that the M06 functionals, corrects for some of it by chance, rather than because the right physics are involved.

A further, more serious concern is that the authors – if I understand correctly – propose to use the same geometry for the high-level (hybrid) calculations obtained from the PBC calculations with PBE+D3. This procedure is also common to compute the electronic structure (see, e.g. <https://doi.org/10.1103/PhysRevB.84.245115>). However, for adsorption energies the approach is questionable. Generally, for the adsorbate/substrate system, the adsorbate will be out of its equilibrium position, i.e. at a too high energy. This systematically underestimates the adsorption energy. In some cases, it may also yield qualitatively incorrect results, as recently shown in <https://doi.org/10.1039/DOCP06605B>. Once again, it is likely that this is not an issue for the small molecules used in this benchmark study, but it certainly does limit the general applicability of the proposed approach.

I would have some additional, minor claims (for example, why did the authors use the D3 vdW-correction scheme, which fails to account for surface polarizabilities, rather than the physically more appropriate D3surf parameterization <https://pubs.acs.org/doi/10.1021/acs.jpcc.9b08824>, which comes at no additional cost? How stable are the results with the choice of the cluster model?), but I feel that at there is no point in nitpicking here.

Overall, I would like to emphasize that despite all my criticism, I think that this is a nice work. It is well written and the benchmarks are certainly useful for the community. However, I don't think that the proposed scheme is innovative or, at least in the way it is presented, generally applicable to (larger) surface adsorbates. I recommend that the authors provide a good estimate of the numerical accuracy of their calculations (probably reporting adsorption energies only to within 1kcal per mol or even less, as appropriate), discuss the existing literature employing hybrid schemes more, and submit the manuscript to a more specialized computational chemistry journal.

Response to Referee's Comments

Referee: 1

"Araujo et al. present an approach to improve the accuracy of DFT calculations of adsorption energies on transition metal surfaces. They combine periodic calculations with GGA functionals with cluster calculations and hybrid functionals. The results are compared with the 39 available experiments (Wellendorf 2015 and Sharada 2019) showing an excellent agreement: MAE 2.3 kcal/mol (0.10 eV). The set includes problematic molecules such as CO, together with physisorbed ones such as C1-C4 hydrocarbons, although other families of reactions were omitted. The authors provide enough information to allow reproducibility. Overall, I have found the Manuscript interesting, yet the content needs to be improved to be fully convincing:"

Comment 1: The way the authors chose the cluster size raises several questions. Figure S2: 7 atoms for H/Ni100 (a typo duplicates Ni111), 8 for CO/Pd100, 9 for H/Ni111, 10 for H/Pt111...First, it was not clear why the cluster size differs for H on Ni111 and Pd111, being both of the same group. I would have expected an unique surface for all fcc(111), another for fcc(100), and yet another for hcp(0001) surfaces.

Response: Since the correction depends weakly on cluster size and shape, but strongly on defining the same electronic spin-coupling at the two levels of calculation on the cluster, we have to put the focus on the electronic structure. We have added text accordingly:

Page 5:

"Two points were considered when building the clusters (Figure S2). Firstly, adsorbates must be coordinated (first shell) similarly to the extended surface calculation. Secondly, the calculation of the term ($E_{HB}^{Ads,Cluster} - E_{PBE+D3}^{Ads,Cluster}$) has to be performed with clusters in the same spin state and coupling for the PBE+D3 and HB calculations. For open *d*-shell metals like Ni it then becomes necessary to limit the cluster size in order to avoid spurious effects of inequivalent electronic structures that become difficult to control for larger cluster sizes. Thus, the focus has been to use clusters that are *electronically similar* at the two levels of calculation rather than using the same cluster size and shape to represent all *structurally similar* surfaces, *e.g.*, Pd(111) and Ni(111), which are rather different in terms of electronic structure. This is based on the fact that, since the correction applies to the local bond and is obtained as the difference between two calculations, it becomes rather insensitive to the cluster size and shape. To demonstrate this, we consider CO adsorption on the top site of Cu(111); using copper eliminates issues around the spin-coupling and allows easy extension to large clusters.

We have used clusters with 7, 10, 13, 17, 20 and 24 Cu atoms (Figure 3). Between the 7- and 24-atoms clusters, we find a difference of 0.5 kcal mol⁻¹ in the correction, in spite of a chemisorption energy difference of 9.2 kcal mol⁻¹. The largest difference in chemisorption energy between any of the clusters is 16.1 kcal mol⁻¹ with a difference in the correction of 2.1 kcal mol⁻¹. There is thus a slight dependence on the cluster size, but the variation is within the error bars of typical experiments (Table 3). We can thus select clusters that are small enough to reliably identify the proper electronic states with reasonable computational effort ensuring robustness to the approach, but still without losing accuracy."

Figure 3: Adsorption energy of CO on Cu clusters with increasing number of atoms using M06 (orange) and PBE+D3 (blue). Moreover, the employed correction to the PBC adsorption energy, $E(\text{M06}) - E(\text{PBE+D3})$, is shown in gray. The variation of the adsorption energy vs. the number of atoms in the Cu clusters with M06 and PBE+D3 is similar, hence, the corrective term becomes almost constant.

Comment 2: Second, and I would like to emphasize this point: The clusters seem exceedingly small. Metal nanoparticles have wild variations in their electronic structure until they reach diameters of around 15\AA or 50+ atoms. Some properties may converge with smaller particles. In Reference 38 (10.1103/PhysRevLett.98.176103, see Figure 2 and the text) an early convergence of 16 atoms is reported. Yet, in the present work much smaller particles having less than half atoms were used.

Response: What the referee points out is well-known and already discussed in the initial submission together with the explanation of this phenomenon. The relevant citations were given. We have expanded the section where this is discussed:

Page 3:

“Using a small metal cluster to correct the interaction energy may seem at odds with the well-known strong variations with cluster size of the computed chemisorption energy. In recent reviews, Hofmann *et al.*²⁶ and Jones *et al.*¹⁹ discuss approaches to obtain reliable energetics both from calculations under PBC and using cluster models, also including extensions based on embedding techniques. These are very valuable and can be made highly accurate for systems dominated by electrostatics or covalent bonding, but for metallic bonding the cluster-size convergence to zero bandgap is slow and erratic. An example is given by the chemisorption energy of hydrogen on Ni(100) that was found to differ by 9 kcal/mol when calculated using clusters of 113 and 118 Ni atoms²⁸. Efforts to overcome this include the self-consistent embedding by Carter and coworkers²¹, but so far only applied to copper. However, copper, as well as silver, has the valence *d*-shell essentially fully occupied and as such does not represent the complexity of transition-metal spin-coupling. Here, we overcome this complexity and

present a systematic and general approach that is applicable to both molecular and dissociative adsorption, as well as to challenging open-*d*-shell transition metals without the need for embedding. In terms of philosophy our approach is equivalent to ONIOM²² with the reference state being the full periodic DFT calculation.”

Pages 3-4:

“The cluster is in reality a small molecule with discrete energy levels, which may or may not include a sizeable band gap. This leads to, *e.g.*, the well-known even-odd alternations in reactivity and other properties in cluster physics. These are direct and real physical effects due to the confinement of the electrons in a finite volume of space (particle in a box) and lead to slow and erratic convergence of the computed chemisorption energy with cluster size²⁸ when computed as the difference between the ground state of the cluster and the chemisorbed system. However, in order for the bond between the surface and adsorbate to form, the electronic structure of both surface and adsorbate must change, which can only be achieved through mixing with excited states. For the extended metal surface, such excitations occur around the Fermi level at zero or small energy cost. For the metal cluster, however, a finite excitation energy may or may not have to be invested to reach the bonding state, and this rehybridization cost reduces the resulting computed chemisorption energy. By calculating this excitation energy for the cluster and adding it to the computed chemisorption energy, surprisingly accurate binding energies can be obtained also from quite small cluster models^{22–25}. Thus, the *local bond* is well defined already for small clusters. However, since the *chemisorption energy* includes the cost to excite the cluster to the bonding state, and this varies strongly with cluster size and shape, the result is the observed strong variations.”

Comment 3: The authors mention databases of 40+ well-defined experimental adsorption energies (Wellendorf 2015 and Sharada 2019), yet only 34 of them were investigated. Why? One wonders why important reactions, such as the dissociative adsorption of CH₃I and CH₂I₂ were omitted (reactions 24, 25, 27 in Wellendorf 2015). The authors seem to have included only the non-dissociative CH₃I adsorptions from Sharada 2019. Including all reactions in the dataset would have shown that the present approach also works for dissociative adsorptions of hydrocarbons, going beyond that of simple diatomic molecules H₂ or O₂.

Response: We have added these cases and confirmed that they, as surmised, are also corrected properly.

Comment 4: Combining periodic GGA with high-level cluster calculations does not seem fully novel when compared to that of Ref 38. The authors may highlight the novelty.

Response: The novelty lies in the extension to open-*d*-shell transition metals in combination with the locality of the surface-adsorbate chemical bond. We acknowledge the similarity in philosophy to the ONIOM approach. See response to Comment 2.

Referee: 2

”The paper reports adsorption energies for a large range of systems calculated with a cluster-corrected approach that is equivalent to the long-standing additive ONIOM scheme (J. Phys. Chem. 1996, 100, 50, 19357–19363) and previously employed by the groups of Sauer, Reuter (Refs. 37, 38), and others not cited such as Carter (e.g. J. Phys. Chem. C 2008, 112, 12, 4649–4657). As such, the approach itself is not novel. The paper presents novel results, namely adsorption energies for a comprehensive range of reactions and reports excellent average errors across the dataset, which comprises equilibrium structures of small chemisorbed and physisorbed adsorbates on various metal surfaces. The main claim is that this approach can predict highly accurate interaction energies by correcting any functional with cluster calculations performed with any desired high-level functional to achieve “arbitrary accuracy”. It is further claimed that the correction is local and should therefore also apply to barriers in reaction sequences. Unfortunately, I do not find the work convincing. The conclusions do not

go significantly beyond the findings of previous works and the manuscript does not provide sufficient information to corroborate the central claims.”

Comment 1: The method stands and falls with the claim that the correction is local. However, this is not explicitly shown. The authors do not describe the rationale for the shape and size of the clusters that are cut from the surfaces and they do not provide quantitative evidence for the swift convergence of $E^{\text{ads}}_{\text{HB}} - E^{\text{ads}}_{\text{PBE+D3}}$ as a function of cluster size.

Response: The locality has been shown in the earlier works that were cited in our submission. We also demonstrate it for the case of CO/Cu(111) in our revised manuscript. See response to Comment 1 of referee 1.

Comment 2: Note that total energy does not converge smoothly and that previous works showed some locality only for differences of E_{xc} . The choice of cluster size therefore appears arbitrary.

Response: It is correct that the total energy does not converge smoothly. This is the reason why bond-preparation (references 22-25 and 27) is necessary if one wants to model a metallic extended surface using a finite-size cluster. However, the correction depends much more weakly on cluster size and shape. We have added Figure 3 with results for CO/Cu(111) (see response to Comment 1 of referee 1).

Comment 3: In general, no systematic convergence data is provided how for example the different type of basis and size of basis set between the local atomic orbital and periodic plane wave code affect the results.

Response: The PBC and cluster calculations of the interaction energies are independent. There is no connection between them in terms of basis set. The cluster calculations have to be performed using the same basis set at both levels. We have added the following:

Page 6:

“We note that using different basis sets in the PBC (plane waves) and cluster (atom-centered Gaussian basis) calculations is a fully consistent procedure since equation (1) only contains adsorption energies computed as an energy difference for each system separately.”

Comment 4: The argument on BSSE is not convincing as higher-level methods typically require larger basis sets with ensuing higher BSSE. If it is negligible then this should be shown. If not, then this and other numerical considerations clearly limit the ability to achieve “arbitrary accuracy” (or rather precision).

Response: The same basis set must be used in the lower- and higher-level calculations. We have expanded the text and added the calculated BSSE values to the SI (Table S4):

Page 6:

“The effects of basis set superposition errors (BSSE) on E_{ads} were evaluated, but found to be similar for PBE+D3 and M06 for the present set of adsorbates and thus cancel out when taking the difference (Table S4). We note that using different basis sets in the PBC (plane waves) and cluster (atom-centered Gaussian basis) calculations is a fully consistent procedure since equation (1) only contains adsorption energies computed as an energy difference for each system separately. However, naturally, the cluster calculations have to be performed using the same basis set for both the lower- (PBE+D3) and higher-level (M06) approaches in order to be comparable.”

Table S4: Basis set superposition error, in kcal mol⁻¹, for randomly selected reactions.

Reaction	PBE+D3-BSSE	M06-BSSE	BSSE(PBE+D3-M06)
C6H6/Pt(111)	-5.54	-6.00	0.46
C6H6/Cu(111)	-2.08	-2.08	0.00
C6H6/Ag(111)	-1.22	-1.31	0.09
C6H6/Au(111)	-3.46	-3.23	-0.23
NO/Pt(111)	-1.38	-0.92	-0.46
CO/Pt(111)	-2.08	-1.85	-0.23
CO/Cu(111)	-1.61	-1.38	-0.23
CO/Ni(111)	-1.15	-0.46	-0.69
CO/Ru(0001)	-1.85	-1.85	0.00
O/Pt(111)	-1.15	-0.92	-0.23

Comment 5: The error analysis is limited to MAEs and RMSE, but tables in the manuscript report MAX errors. It would be interesting to provide a deeper analysis what structures are associated with these maximum errors and if these structures are the same that provide large errors for conventional functionals. Is the structure of the error distribution of e.g. PBE preserved after correction or is the remaining error of the scheme random/statistical?

Response: We have added the following text to the manuscript to elaborate on the reported MAX:

Page 12:

“The largest error ($|MAX|$) displayed by the PBE+D3/M06 is 7.6 kcal mol⁻¹ for the CH₃I+Pt(111)→ CH₃/Pt(111) reaction (Table 1). The functional best describing the adsorption of CH₃I is the PBE+D3 with an error of only 1.6 kcal mol⁻¹, thus, for this case, the corrective scheme does not improve the description. For most of the functionals, $|MAX|$ corresponds to the adsorption of N on Ni(100) (with the exception of GAM that has $|MAX|$ for benzene adsorption on Pt(111)) with values ranging from 23.4 kcal mol⁻¹ for BEEF-vdW up to 48.6 kcal mol⁻¹ for SCAN. In general, the corrective scheme results in lower ($|MAX|$) than the other approaches displayed in Table 1, which highlights the predictive power of the approach.”

Comment 4: For anything else than late and noble transition metal clusters, the question of the spin and electronic state of the cluster will become increasingly difficult with a high level of manual intervention. It is not clear from the manuscript how this will be treated for more challenging systems and for nonequilibrium structures (transition states etc). To some extent, the discussion on page 13 around benzene already shows the limitations of this approach.

Response: The discussion around benzene was necessary since earlier works had not considered the possibility of not having the correct structure. Applying the concept of capitalistic chemistry shows that the $\pi \rightarrow \pi^*$ excitation can actually give a return on the rehybridization/excitation investment that is larger than for the structure assumed in previous benchmarks. As we see it this example supports our approach, although we agree that it makes the calculations less “standard”.

Comment 5: I was not convinced by the qualitative argument around capitalistic chemistry. This relates to penalties due to electronic and geometrical preparation to adsorption, which is not even analysed in this manuscript but only relates to discussions in other works. The argument is that GGAs get the local chemistry, preparation penalty etc. wrong and that those will be the same on cluster and PBC and therefore can be corrected by subtraction? This should be explored in more detail.

Response: This has actually been explored in some detail in the cited references. Note that the preparation penalty on the adsorbate is not the issue here, neither that on the cluster. It is rather used to underline that the origin of the chemisorption energy variations with cluster size is the

specific electronic structure of the various cluster models and, when that is taken into account, stable chemisorption energies can be extracted from clusters quite independent of size and shape.

Comment 6: The authors raise the context of computing accurate barriers, but I am not convinced that this approach will provide as accurate barriers as it does adsorption energies or that it will provide smooth and sensible potential energy landscapes. This should be systematically studied. Transition states (non-equilibrium states in general) exhibit more non-local chemistry than ground states, so it would be important to know if the same arguments hold.

Response: The approach corrects interaction energies without assumptions on structure. We agree that cluster models may have to be extended, but the approach will be the same. We have added a final sentence:

Page 17:

“Furthermore, since the correction works on the interaction energy between metal and adsorbate, we expect it to be equally applicable to verify or correct computed barriers in a reaction sequence, where high accuracy is particularly essential for reliable microkinetic modeling of rates and turn-over frequencies. In the latter case the size of the cluster model may need to be extended as required which may make it more challenging to define a common reference electronic state for the correction.”

Referee: 3

I am reviewing the manuscript of Araujo et al for publication in Nature Communication. The work aims at providing a computational recipe for an accurate prediction of adsorption energies on the surface of transition metals, applying a concept they term “capitalistic chemistry”. The target of the work, obtaining highly accurate adsorption energies throughout chemical space, independent of the type of the bonding, is highly important and an important goal for various communities, including surface science, organic electronics, catalysis, and many more. Unfortunately, for the reasons discussed below, I am unable to recommend the publication of this work in the present form.

Comment 1: The analogy with economics and business decisions seems intriguing and appealing at first. However, throughout the work it becomes clear that this frame is just a re-wording of well-known chemical concepts, unnecessarily introducing jargon. The analogy also seems to be incorrectly used sometimes. For example, the authors define the cost to re-hybridize as energetic investment (with “energy” as unit), and the energetic gain through the formation of the new bond as “return on investment” (which would also have energy as unit). However, while there are various possible definitions on ROIs, it is customary to define it as the ratio between net income and investment (this being unitless). Also, it is usually employed to decide whether a certain investment should be undertaken (when given multiple opportunities), which is where the analogy, again, falls short. While this could be solved by using more appropriate terms, I feel that mixing the jargons of chemistry and economics does not provide any additional insight if it doesn’t introduce a new, beneficial concept. It therefore should be removed completely.

Response: We have endeavored to better clarify the concept and also how it can be used to investigate multiple possibilities. The use of the term “return on investment” has been modified.

Page 4:

“This can be simply summarized in terms of the concept of “capitalistic chemistry”^{29,30}: The energy cost to rehybridize the electronic (and geometric) structure of the reactants is the *investment*. The difference between the energy gained through the formation of the new bond (bond strength) and the investment is then the *profit*, *i.e.* the chemisorption energy. This concept”

thus directly shows that it can be misleading to use the magnitude of the resulting energy as a measure of the strength of an interaction or to distinguish between chemisorption and physisorption. A small value can be due to a weak interaction or to a strong bond associated with a high rehybridization cost.

“Capitalistic chemistry” is also predictive in terms of possible bonding schemes of an adsorbate to a surface. A σ -bond between a hydrocarbon and Pt (in Pt(111)) has been estimated to 53 kcal/mol by Carter and Koehl³¹ and used to distinguish between reaction mechanisms in the decomposition of ethylene on Pt(111) by comparing the energy cost to reach the involved adsorbate excited states in the gas phase with the gain from bond formation. A similar estimate of ~ 50 kcal mol⁻¹ was obtained for unsaturated hydrocarbons on Cu surfaces by Triguero *et al.*²⁹ by explicitly taking into account the $\pi \rightarrow \pi^*$ excitation energy to reach the bond-prepared di-radical state that can form the two σ -bonds required for a lying-down geometry. For CO and N₂, the cost for this excitation is too high to be offset by two σ -bonds, resulting in vertical chemisorption unless additional interactions are available^{2,32}. Here, we use this concept to remind the reader why the *local* chemical bond can be well described by cluster models^{22–25}, while the chemisorption energy requires special consideration. In addition, we apply it to the cases of chemisorption of benzene and CH on Pt(111).”

Page 14:

“Indeed, we find a reconstruction from three-fold to four-fold hollow of the Pt atoms interacting with the adsorbate and the quinoid structure was retained also in the PBC calculation (see SI for the C₆H₆/Pt(111) structure). This structure gave an adsorption energy of -53.2 kcal mol⁻¹ with the PBC-PBE+D3 approach while the adsorption energy computed in the standard way (four layers Pt with no preoptimization of the C₆H₆ on the finite cluster) resulted in -47.0 kcal mol⁻¹. This difference is mainly due to the added flexibility in the slab model with more layers allowed to move -- as indicated by the cluster calculation -- that allowed the more stable structure to be found. Very interestingly, after the correction, the error coming from the 5 layers slab model for the quinoid state is 4.8 kcal mol⁻¹ while for the four-layer slab, resulting in rather undistorted planar benzene, the error even after M06 correction is 11.1 kcal mol⁻¹. The combination of PBC and cluster calculations thus allowed the flexibility to find the proper chemisorbed structure for this system. The search for this structure was guided by the expected return on the rehybridization investment, which could be estimated as (2 σ -bonds $\sim 2 \times 53$ kcal/mol)/90 kcal/mol if the necessary two σ -bonds could form³¹.”

Page 15:

“This can be simply understood in terms of the concept of “capitalistic chemistry”^{29,30} where any chemical reaction must involve excited states in order to break and form new chemical bonds. The required excitation costs energy, which may be regarded as the investment necessary to reach the bonding state. Forming the new bond releases energy and the difference between the rehybridization investment and the energy released is then the resulting chemisorption energy, or profit. For the extended metal surface such excitations occur around the Fermi level at zero or low energy (investment) cost, but for a small cluster the excitation energy will vary with cluster size.”

Comment 2: On more scientific grounds, the authors suggest a hybrid scheme of low-level band-structure and high-level cluster calculations. Such QM/QM schemes that add and subtract energies at different levels of theory are a common concept for interface calculations(see, e.g. <https://pubs.rsc.org/en/content/articlelanding/2013/CP/c3cp52321g#!divAbstract>). I am afraid that I fail to see the innovative character of the proposed scheme and why it would be published in Nature Communications. Even if there is an innovative aspect that I am missing, I feel that a comprehensive discussion of earlier works with such hybrid and/or other embedding schemes is missing.

Response: We have included a brief discussion of other schemes and how our approach differs:

Page 3:

“Hu *et al.*¹⁷ have shown that this approach can predict the site where CO adsorbs on Cu(111) by using the hybrid functional B3LYP for the correction. Using a small metal cluster to correct the interaction energy may seem at odds with the well-known strong variations with cluster size of the computed chemisorption energy. In recent reviews, Hofmann *et al.*²⁶ and Jones *et al.*¹⁹ discuss approaches to obtain reliable energetics both from calculations under PBC and using cluster models, also including extensions based on embedding techniques. These are very valuable and can be made highly accurate for systems dominated by electrostatics or covalent bonding, but for metallic bonding the cluster-size convergence to zero bandgap is slow and erratic. An example is given by the chemisorption energy of hydrogen on Ni(100) that was found to differ by 9 kcal/mol when calculated using clusters of 113 and 118 Ni atoms²⁸. Efforts to overcome this include the self-consistent embedding by Carter and coworkers²¹, but so far only applied to copper. However, copper, as well as silver, has the valence *d*-shell essentially fully occupied and as such does not represent the complexity of transition-metal spin-coupling. Here, we overcome this complexity and present a systematic and general approach that is applicable to both molecular and dissociative adsorption, as well as to challenging open-*d*-shell transition metals, without the need for embedding. In terms of philosophy our approach is equivalent to ONIOM²² with the reference state being the full periodic DFT calculation.”

Comment 3: Without the innovative character of a hybrid scheme, the present work is mostly a benchmark paper for the performance of various computational schemes for the adsorption of small molecules on transition metals. Such a work is clearly important. Unfortunately, there are also some more technical issues. In order to provide “accurate and balanced” adsorption energies with the given hybrid schemes (and in order to transfer the findings to other systems), it is of paramount importance that the individual terms of eq. 1 are numerically well converged with respect to the computational settings and, therefore, robust. The authors have mostly used 4-layer slabs throughout the work. I doubt that, even for small molecules, this yields energies that are converged to less than 0.1 kcal/mol. Likewise, also a 4x4 k-point grid would be reasonable for a standard investigation, but I am not convinced that it is converged to the 0.1 kcal/mol accuracy required to make the claims in that paper. In fact, already the employed broadening scheme in the PBC calculations, 2nd-order MP, may be at the verge of having a too large broadening parameter (0.2 eV) to allow for a sufficiently faithful backextrapolation to zero-broadening (see also <https://doi.org/10.1039/D0CP06605B>) with that accuracy.

Response: We have repeated all calculations with an 8x8x1 k-point grid and 700 eV cutoff (previously 4x4x1 and 500 eV). This resulted in a change of 0.2 kcal/mol in MAE and RMSE to 2.4 and 2.9 kcal/mol compared to 2.2 and 2.7 kcal/mol previously. It should be noted that we also added 4 more reactions to the data set (dissociative chemisorption of CH₃I and CH₂I₂, see comment 3 by referee 1).

We have added a brief comment on the use of four-layer slabs:

Page 6:

“A four-layered slab was used to model the metal surfaces where the two topmost layers were allowed to optimize while the two bottom layers were fixed at the optimized bulk structure; this is the same approach as in previous benchmark studies.”

We do not claim an accuracy of 0.1 kcal/mol as is evident from our reported MAE and RMSE values.

We do discuss experimental uncertainties (average: 1.6 kcal/mol) that should be taken into account when discussing benchmarks.

We are of course aware that the calculations can be improved, but believe that the most significant aspect would be to address the wave function treatment, as we discuss in the final paragraph of the introduction.

Comment 4: Furthermore, I am concerned about the author's statement that the BSSE between PBE+D3 and HB cancels out. There is no fundamental reason why it should, and in fact, it is well known that more correlated methods show a larger BSSE than semilocal functionals. I would expect that the authors find this cancellation here only because the molecules they investigated were very small; I very much doubt that it will hold for larger systems (where the BSSE becomes larger in absolute terms).

Response: We have added Table S4 to the SI with our computed values. We do not find the suggested larger BSSE for M06:

Table S4: Basis set superposition error, in kcal mol⁻¹, for randomly selected reactions.

Reaction	PBE+D3-BSSE	M06-BSSE	BSSE(PBE+D3-M06)
C6H6/Pt(111)	-5.54	-6.00	0.46
C6H6/Cu(111)	-2.08	-2.08	0.00
C6H6/Ag(111)	-1.22	-1.31	0.09
C6H6/Au(111)	-3.46	-3.23	-0.23
NO/Pt111	-1.38	-0.92	-0.46
CO/Pt(111)	-2.08	-1.85	-0.23
CO/Cu(111)	-1.61	-1.38	-0.23
CO/Ni(111)	-1.15	-0.46	-0.69
CO/Ru(0001)	-1.85	-1.85	0.00
O/Pt(111)	-1.15	-0.92	-0.23

Comment 5: Along the same lines, I am concerned about the use of different basis sets for PBC and cluster calculations. I understand that, to some extent, this happens out of necessity. Not many codes can deal with both clusters and PBCs with the same basis set, but some can. However, the different limitations of the two basis sets (plane wave with 500 eV cutoff in a given unit cell versus LCAO triple-zeta) is likely to introduce a certain systematic bias in the benchmark. I would suspect that the M06 functionals, corrects for some of it by chance, rather than because the right physics are involved.

Response: We have increased the plane wave cutoff to 700 eV without significant effects (0.2 kcal/mol – see Comment 3; all values have been updated). We have added a comment on the use of different basis sets in the PBC and cluster calculations:

Page 6:

"We note that using different basis sets in the PBC (plane waves) and cluster (atom-centered Gaussian basis) calculations is a fully consistent procedure since equation (1) only contains adsorption energies computed as an energy difference for each system separately. However, naturally, the cluster calculations have to be performed using the same basis set for both the lower- (PBE+D3) and higher-level (M06) approaches in order to be comparable."

Comment 6: A further, more serious concern is that the authors – if I understand correctly – propose to use the same geometry for the high-level (hybrid) calculations obtained from the PBC calculations with PBE+D3. This procedure is also common to compute the electronic structure (see, e.g. <https://doi.org/10.1103/PhysRevB.84.245115>). However, for adsorption energies the approach is questionable. Generally, for the adsorbate/substrate system, the adsorbate will be out of its equilibrium position, i.e. at a too high energy. This systematically underestimates the adsorption energy. In some cases, it may also yield qualitatively incorrect results, as recently shown in <https://doi.org/10.1039/D0CP06605B>. Once again, it is likely that

this is not an issue for the small molecules used in this benchmark study, but it certainly does limit the general applicability of the proposed approach.

Response: The aim is to correct the interaction at the given point of the PES. We have added a comment:

Page 6:

“We comment briefly on using the same geometry for the PBE+D3 and M06 calculations on the cluster. This is intentional since the M06 is used to obtain an improved estimate of the interaction at the geometry of the PBC calculation. Reoptimizing the adsorbate on the small cluster, even with the cluster atoms fixed, may introduce uncertainties in terms of, *e.g.*, edge effects and lose the connection with the optimized structure in the PBC calculation, which typically includes effects of coverage and coadsorbates.”

Comment 7: I would have some additional, minor claims (for example, why did the authors use the D3 vdW-correction scheme, which fails to account for surface polarizabilities, rather than the physically more appropriate D3surf parameterization <https://pubs.acs.org/doi/10.1021/acs.jpcc.9b08824>, which comes at no additional cost? How stable are the results with the choice of the cluster model?), but I feel that at there is no point in nitpicking here.

Response: Concerning D3^{Surf} we have added the following:

Page 6:

“An improved (D3^{Surf}) approach that reduces the polarizability for highly-coordinated atoms has recently been proposed, but so far with parameters only for Cu and Ag. [Ref *J. Phys. Chem. C* 2019, 123, 48, 29219–29230].”

Comment 8: Overall, I would like to emphasize that despite all my criticism, I think that this is a nice work. It is well written and the benchmarks are certainly useful for the community. However, I don't think that the proposed scheme is innovative or, at least in the way it is presented, generally applicable to (larger) surface adsorbates. I recommend that the authors provide a good estimate of the numerical accuracy of their calculations (probably reporting adsorption energies only to within 1kcal per mol or even less, as appropriate), discuss the existing literature employing hybrid schemes more, and submit the manuscript to a more specialized computational chemistry journal.

Response: We appreciate the positive remark and believe that we have provided all the explanations necessary for our work to now be accepted for publication.

Reviewers' comments:

Reviewer #1 (Remarks to the Author):

Araujo et al. improved the accuracy of adsorption on extended metal surfaces, done in periodic boundary conditions (PBE), by recalibrating the "local" part of the bond with two cluster calculations with one done at higher levels of theory. The correction is reasonably converged for clusters as small as 13 atoms (and perhaps 7 atoms, error bars in the order 2 kcal mol^{-1} or 0.10 eV). Reasons behind the selection of a particular cluster size (such as Ni₁₁₁, which differs from Pd₁₁₁) were rationalized in terms of controlling the spin state. The discussion of the electronic structure was strengthened.

The authors made an effort to reply satisfactorily to all concerns raised by the Reviewers (most of them shared). Comment 2.6 is perhaps an exception. In my understanding, Reviewer #2 was not convinced that the method provided by the authors to get adsorption energies, can be extended to transition states expecting similar accuracies. In other words, the claim "we expect it to be equally applicable to verify or correct computed barriers in a reaction sequence" (line 598) does not sit on solid ground as "Transition states (non-equilibrium states in general) exhibit more non-local chemistry than ground states" (Reviewer 2). The authors replied by commenting that the cluster size "may need to be extended". That is more of a procedural note, rather than assessing if the procedure will keep its accuracy. The authors would better rephrase that statement, or support it with hard evidence (As they did with Figure 3) and avoid ambiguity.

Minor comments, also regarding ambiguity:

* Last sentence in the abstract needs to be refined (Line 18-19). Does "reaction sequence" mean a mechanism containing a series of reactions? Is the term "accuracy" related here to reaction or activation energies? When first reading the abstract I got the impression they were "reaction energies". Yet they mutate to "activation energies" in Line 599. Activation energies are also discussed on lines 159 and 355 without further support.

* "However, since the chemisorption energy includes the cost to excite the cluster to the bonding state, and this varies strongly with cluster size and shape, the result is the observed strong variations." Lines 131-133

* "A small value can be due to [eg. A weak binding energy can be caused either by?] a weak interaction or to a strong bond associated with a high rehybridization cost." Lines 139-140.

Reviewer #2 (Remarks to the Author):

The authors have addressed some of my points and have provided additional data to the manuscript.

The authors have not addressed my comments regarding activation energies and energy landscapes. (point 6)

The main impact and motivation of the manuscript stem from its prospective use in catalysis to predict accurate activation barriers. The authors specifically state this in the conclusions. Yet the manuscript does not provide evidence that this approach also works for transition states or that it delivers smooth energy paths along reaction coordinates. The authors suggest that this is possible with this approach without any data to corroborate this claim. This should be systematically studied. Transition states (non-equilibrium states in general) exhibit more non-local chemistry than groundstates, so it would be important to know if the same arguments hold.

Reviewer #3 (Remarks to the Author):

In their revision, the authors have addressed most of the concerns of the referees, some satisfactorily, others less so.

My main issue is still stems from the use of “capitalistic chemistry”: A reaction is exothermic if the cost of breaking bonds is less than gain of forming new bonds. This is a basic concept chemistry. The authors rephrase this (and I paraphrase here) as a reaction being profitable if the investment is less than the return. The introduction of this jargon (it isn't a new concept, in my opinion) is not helpful. Quite the contrary, it is often confusing and sometimes misleading.

Stripping away this framework, what remains is a QM/QM-approach that is similar to previous works (like the ONIOM-approach, as the authors themselves acknowledges). There is little truly novel in this approach. In this assessment, all three referees seem to concur. The authors claim that the novelty lies in being able to deal with open-d-shell-systems, but I do not feel that is enough of an advancement to justify publication in Nature Communications.

I realize that the how “new” or important an advancement is lies clearly in the eye of the beholder. It is, of course, the editors prerogative to ignore my opinion here.

Nonetheless, I also still have scientific concerns. This is best summarized via one the authors rebuttals:

“We are of course aware that the calculations can be improved, but believe that the most significant aspect would be to address the wave function treatment, as we discuss in the final paragraph of the introduction.”

I regret to say that I do not agree with this statement or the philosophy behind this. Improving the wave function treatment is only necessary or sensible if it is the largest source of errors in the calculations. I am not convinced that this the case here. More importantly, however, if benchmarks are reported, it is imperative that all other sources of errors are below the reported significance. Otherwise, the improvement due to a better method is mixed with the numerical errors that incur, making the whole assessment somewhat misleading. Although I appreciate that the authors have gone a long way to improve their quality, I am still unconvinced that the results are sufficiently converged to report MAEs with a sub-kcal/mol significance.

REVIEWER COMMENTS AND RESPONSE

Reviewer #1 (Remarks to the Author):

Araujo et al. improved the accuracy of adsorption on extended metal surfaces, done in periodic boundary conditions (PBE), by recalibrating the "local" part of the bond with two cluster calculations with one done at higher levels of theory. The correction is reasonably converged for clusters as small as 13 atoms (and perhaps 7 atoms, error bars in the order 2 kcal mol⁻¹ or 0.10 eV). Reasons behind the selection of a particular cluster size (such as Ni₁₁₁, which differs from Pd₁₁₁) were rationalized in terms of controlling the spin state. The discussion of the electronic structure was strengthened.

Comment:

The authors made an effort to reply satisfactorily to all concerns raised by the Reviewers (most of them shared). Comment 2.6 is perhaps an exception. In my understanding, Reviewer #2 was not convinced that the method provided by the authors to get adsorption energies, can be extended to transition states expecting similar accuracies. In other words, the claim "we expect it to be equally applicable to verify or correct computed barriers in a reaction sequence" (line 598) does not sit on solid ground as "Transition states (non-equilibrium states in general) exhibit more non-local chemistry than ground states" (Reviewer 2). The authors replied by commenting that the cluster size "may need to be extended". That is more of a procedural note, rather than assessing if the procedure will keep its accuracy. The authors would better rephrase that statement, or support it with hard evidence (As they did with Figure 3) and avoid ambiguity.

Response:

We have now performed the requested calculations for transition states and demonstrate that the procedure is applicable also in this case.

Minor comments, also regarding ambiguity:

Comment:

* Last sentence in the abstract needs to be refined (Line 18-19). Does "reaction sequence" mean a mechanism containing a series of reactions? Is the term "accuracy" related here to reaction or activation energies? When first reading the abstract I got the impression they were "reaction energies". Yet they mutate to "activation energies" in Line 599. Activation energies are also discussed on lines 159 and 355 without further support.

Response:

We have rewritten the abstract to remove the ambiguity and conform to the 150 word limit. Activation energies are now used specifically for the computed barriers.

Comment:

* "However, since the chemisorption energy includes the cost to excite the cluster to the bonding state, and this varies strongly with cluster size and shape, the result is the observed strong variations." Lines 131-133

Response:

We have tried to clarify by writing more explicitly (Summary and Conclusions):

"However, it has been shown that, if one takes into account the excitation energy needed to reach the bonding state of the specific cluster, then very stable and reliable chemisorption energies can be obtained also from small cluster models of the metal surface²⁴⁻²⁸. This can be simply understood in terms of the concept of "capitalistic chemistry"^{29,30} which allows to distinguish between bond strength

and reaction energy by considering the excited states that must be involved in order to break and form new chemical bonds.

The change in electronic structure from involving excited states costs energy, which may be regarded as the investment necessary to reach the bonding state. Forming the new bond releases energy, which corresponds to the return on the investment. The difference between the rehybridization investment and the energy released is then the resulting chemisorption energy, or profit; knowledge of the excited energy levels of a given molecule thus allows predicting its possible binding modes^{2,29}. For the extended metal surface such excitations occur around the Fermi level at zero or low energy (investment) cost, but for a small cluster the excitation energy will vary with cluster size."

Comment:

* "A small value can be due to [eg. A weak binding energy can be caused either by?] a weak interaction or to a strong bond associated with a high rehybridization cost." Lines 139-140.

Response:

We have removed this statement.

Reviewer #2 (Remarks to the Author):

The authors have addressed some of my points and have provided additional data to the manuscript.

The authors have not addressed my comments regarding activation energies and energy landscapes. (point 6)

Comment:

The main impact and motivation of the manuscript stem from its prospective use in catalysis to predict accurate activation barriers. The authors specifically state this in the conclusions. Yet the manuscript does not provide evidence that this approach also works for transition states or that it delivers smooth energy paths along reaction coordinates. The authors suggest that this is possible with this approach without any data to corroborate this claim. This should be systematically studied. Transition states (non-equilibrium states in general) exhibit more non-local chemistry than groundstates, so it would be important to know if the same arguments hold.

Response:

We have now performed the requested calculations for transition states and demonstrate that the procedure is applicable also in this case.

Reviewer #3 (Remarks to the Author):

In their revision, the authors have addressed most of the concerns of the referees, some satisfactorily, others less so.

Comment:

My main issue is still stems from the use of "capitalistic chemistry": A reaction is exothermic if the cost of breaking bonds is less than gain of forming new bonds. This is a basic concept chemistry. The authors rephrase this (and I paraphrase here) as a reaction being profitable if the investment is less than the return. The introduction of this jargon (it isn't a new concept, in my opinion) is not helpful. Quite the contrary, it is often confusing and sometimes misleading.

Response:

The reviewer is of course right about the definition of exothermicity. The usefulness of the concept stems rather from its predictive capability, i.e. knowledge of the available excited states of a molecule allows to determine whether a specific binding mode will be exothermic or not, as well as to extract the strength of an interaction independent of the resulting exothermicity. We have attempted to emphasize this aspect better:

Abstract:

"To justify the use of small clusters we introduce the concept of "capitalistic chemistry" and exemplify by applying to benzene on Pt(111)."

Discussion and Conclusions:

This can be simply understood in terms of the concept of "capitalistic chemistry"^{29,30} which allows to distinguish between bond strength and reaction energy by considering the excited states that must be involved in order to break and form new chemical bonds.

The difference between the rehybridization investment and the energy released is then the resulting chemisorption energy, or profit; knowledge of the excited energy levels of a given molecule thus allows predicting its possible binding modes^{2,29}.

Comment:

Stripping away this framework, what remains is a QM/QM-approach that is similar to previous works (like the ONIOM-approach, as the authors themselves acknowledges). There is little truly novel in this approach. In this assessment, all three referees seem to concur. The authors claim that the novelty lies in being able to deal with open-d-shell-systems, but I do not feel that is enough of an advancement to justify publication in Nature Communications.

I realize that the how "new" or important an advancement is lies clearly in the eye of the beholder. It is, of course, the editors prerogative to ignore my opinion here.

Response:

Our work provides a true benchmarking procedure by more formally separating the effects of band structure and coverage - best described by DFT - from that of the local chemical bond for which accurate quantum chemical methods are available. The philosophy has been applied to individual cases before (CO/MgO(100), CO/Cu(111)) as we note, but not presented as a systematic approach. We furthermore note that, based on the original comments from the reviewers, the usefulness of small metal clusters to extract the local bond strength needs to be reemphasized.

We have rewritten the abstract to underline this aspect:

Abstract:

For high accuracy, effects of band structure and coverage, as well as the local bond strength in both covalent and non-covalent interactions, must be reliably described and much focus has been put on improving functionals to this end. Here, we instead use a correction from higher-level calculations on small metal clusters to improve periodic band structure adsorption energies and barriers.

Comment:

Nonetheless, I also still have scientific concerns. This is best summarized via one the authors rebuttals:

"We are of course aware that the calculations can be improved, but believe that the most significant aspect would be to address the wave function treatment, as we discuss in the final paragraph of the

introduction.”

I regret to say that I do not agree with this statement or the philosophy behind this. Improving the wave function treatment is only necessary or sensible if it is the largest source of errors in the calculations. I am not convinced that this the case here. More importantly, however, if benchmarks are reported, it is imperative that all other sources of errors are below the reported significance. Otherwise, the improvement due to a better method is mixed with the numerical errors that incur, making the whole assessment somewhat misleading. Although I appreciate that the authors have gone a long way to improve their quality, I am still unconvinced that the results are sufficiently converged to report MAEs with a sub-kcal/mol significance.

Response:

We report our values with one decimal, which seems rather standard. We have investigated the basis set superposition errors for the cluster calculations and found that they essentially cancel out in the subtraction between the two calculations (lower- and higher-level DFT). We have endeavored to converge the reported PBC calculations. The only weakness we see and recognize lies in the use of an approximate (albeit significantly more reliable) functional in the cluster calculations. Here, we point to possible future improvements in terms of better methods, such as multiconfigurational DFT or even CCSD(T) or RASPT2 that today are out of reach for open-shell transition metal clusters.

Reviewers' comments:

Reviewer #1 (Remarks to the Author):

The Manuscript of Araujo et al reflects that they addressed all technical criticisms raised by the reviewers and has merits publication in Nature Communications.

I only suggest two minor updates:

Lines 106-108 add a reference about the even-odd alternation. Even an odd one. The readership of Nature Communications is very broad. Many readers may not be aware of the quirkiness of small nanoclusters.

Line 526 replace "deeper adsorption energies (more negative)" by "more exothermic".

Reviewer #2 (Remarks to the Author):

The authors have added data on the performance for transition states and the results look promising! This is now round three of the review and all three reviewers remain, somewhat unenthusiastic about the manuscript. Upon rereading the manuscript and SI and the comments of the other reviewers (as well as my own original comments as it has been a while since they were written), I am ready to make a few observations, with which I will finally conclude my contribution to these proceedings.

- All three reviewers have raised the issue of the discussion around capitalistic chemistry and the confusing notion of discussing the role of excited states (in the context of bond preparation) in this work. The edits that the authors have suggested in response to the latest comments of #R1 and #R3 have made the text even more confusing in my opinion. This whole discussion feels somewhat detached from what the manuscript actually deals with. I think it is important to consider the manuscript from the perspective of a first-time reader (which arguably is hard for everyone involved at this stage) who is unfamiliar with the references that are mentioned when discussing capitalistic chemistry.

- Several of the original issues persist. The manuscript and SI as they stand address the issue of cluster size and shape convergence in text and by virtue of references and the results are convincing. Yet, no data on cluster size and shape dependence of energies is presented to alleviate the concerns that readers might have on such a critical issue. On the subject, no recent references are cited. Would it not be better to provide some form of data to eliminate any concerns that might arise?

- I agree with reviewer 3 regarding the novelty of the approach (although I do acknowledge the potential value of this technical correction procedure and the meticulous benchmarking exercise that the authors have conducted). I don't think that a cluster correction of the adsorption energy can be considered a "formal separation of effects of band structure and coverage". While it technically achieves something to that effect, it is not formal in the sense of formal theories. It appears to me that the final comment of #R3 remains a sticking point. Numerical convergence of cluster calculations is also an issue that I have raised in my original comments.

- Finally, the structural XYZ data pasted into the SI PDF should rather be uploaded to a data repository as input files and shared via a DOI. There are also excellent electronic structure repositories that can host both input and output files of standard software packages (e.g. the Molssi QCArchive or the NOMAD repository)

REVIEWER COMMENTS

Reviewer #1 (Remarks to the Author):

The Manuscript of Araujo et al reflects that they addressed all technical criticisms raised by the reviewers and has merits publication in Nature Communications.

I only suggest two minor updates:

Comment 1:

Lines 106-108 add a reference about the even-odd alternation. Even an odd one. The readership of Nature Communications is very broad. Many readers may not be aware of the quirkiness of small nanoclusters.

Response:

We have expanded the discussion (page 3) and include two reviews:

"Using a small metal cluster to correct the interaction energy may seem at odds with the well-known strong variations with cluster size of the computed chemisorption energy. In recent reviews, Hofmann *et al.*²¹ and Jones *et al.*²² discuss approaches to obtain reliable energetics both from calculations under PBC and using cluster models, also including extensions based on embedding techniques. These are very valuable and can be made highly accurate for systems dominated by electrostatics or covalent bonding, but for metallic bonding the cluster-size convergence to zero bandgap is slow and erratic."

Comment 2:

Line 526 replace "deeper adsorption energies (more negative)" by "more exothermic".

Response:

Done!

Reviewer #2 (Remarks to the Author):

The authors have added data on the performance for transition states and the results look promising! This is now round three of the review and all three reviewers remain, somewhat unenthusiastic about the manuscript. Upon rereading the manuscript and SI and the comments of the other reviewers (as well as my own original comments as it has been a while since they were written), I am ready to make a few observations, with which I will finally conclude my contribution to these proceedings.

Comment 1:

- All three reviewers have raised the issue of the discussion around capitalistic chemistry and the confusing notion of discussing the role of excited states (in the context of bond preparation) in this work. The edits that the authors have suggested in response to the latest comments of #R1 and #R3 have made the text even more confusing in my opinion. This whole discussion feels somewhat detached from what the manuscript actually deals

with. I think it is important to consider the manuscript from the perspective of a first-time reader (which arguably is hard for everyone involved at this stage) who is unfamiliar with the references that are mentioned when discussing capitalistic chemistry.

Response:

We have removed all mention of capitalistic chemistry in the revised manuscript. Clearly, a more focused separate presentation is necessary to get this concept across.

Comment 2:

- Several of the original issues persist. The manuscript and SI as they stand address the issue of cluster size and shape convergence in text and by virtue of references and the results are convincing. Yet, no data on cluster size and shape dependence of energies is presented to alleviate the concerns that readers might have on such a critical issue. On the subject, no recent references are cited. Would it not be better to provide some form of data to eliminate any concerns that might arise?

Response:

This information was included in an earlier revision, but we unfortunately made a mistake when revising our manuscript and used a version where this was not included. We have reinstated this discussion (and figure 3) in the present version:

“Thus, the focus has been to use clusters that are *electronically similar* at the two levels of calculation rather than using the same cluster size and shape to represent all *structurally similar* surfaces, e.g., Pd(111) and Ni(111), which are rather different in terms of electronic structure. This is based on the fact that, since the correction applies to the local bond and is obtained as the difference between two calculations, it becomes rather insensitive to the cluster size and shape. To demonstrate this, we consider CO adsorption on the top site of Cu(111); using copper eliminates issues around the spin-coupling and allows easy extension to large clusters.

Figure 3: Adsorption energy of CO on Cu clusters with increasing number of atoms using M06 (orange) and PBE+D3 (blue). Moreover, the employed correction to the PBC adsorption energy, $E(\text{M06})-E(\text{PBE+D3})$, is shown in gray. The variation of the adsorption energy vs. the number of atoms in the Cu clusters with M06 and PBE+D3 is similar, hence, the corrective term becomes almost constant.

We have used clusters with 7, 10, 13, 17, 20 and 24 Cu atoms to represent Cu(111) (Figure 3). Between the 7- and 24-atoms clusters, we find a difference of $0.5 \text{ kcal mol}^{-1}$ in the correction, in spite of a chemisorption energy difference of $9.2 \text{ kcal mol}^{-1}$. The largest difference in chemisorption energy between any of the clusters is $16.1 \text{ kcal mol}^{-1}$, with a difference in the correction of $2.1 \text{ kcal mol}^{-1}$. There is thus a slight dependence on the cluster size, but the variation is within the error bars of typical experiments (Table 3). We can thus select clusters that are small enough to reliably identify the proper electronic states with reasonable computational effort ensuring robustness to the approach, but still without losing accuracy.”

Comment 3:

- I agree with reviewer 3 regarding the novelty of the approach (although I do acknowledge the potential value of this technical correction procedure and the meticulous benchmarking exercise that the authors have conducted). I don't think that a cluster correction of the adsorption energy can be considered a "formal separation of effects of band structure and coverage". While it technically achieves something to that effect, it is not formal in the sense of formal theories. It appears to me that the final comment of #R3 remains a sticking point. Numerical convergence of cluster calculations is also an issue that I have raised in my original comments.

Response:

The numerical convergence and origin of the variations in extracted chemisorption energy have been discussed in earlier work in terms of bond-preparation (refs. 30-34) and is directly related to the variation in required excitation of the clusters to reach the bonding state. We have expanded on this (page 3 and response to comment 1 of referee 1). Furthermore, the relevant quantity is the convergence of the correction which we illustrate in the added Figure 3 (see response to comment 2 above).

Comment 4:

- Finally, the structural XYZ data pasted into the SI PDF should rather be uploaded to a data repository as input files and shared via a DOI. There are also excellent electronic structure repositories that can host both input and output files of standard software packages (e.g. the Molssi QCArchive or the NOMAD repository)

Response:

This is an excellent suggestion that we will take into account in future work, i.e. collect input files in the appropriate format as the project proceeds. In the present case we feel that too much time has passed since preparing the original SI PDF that going back to the various computers that were used to extract the correct files might introduce errors.

Reviewers' comments:

Reviewer #1 (Remarks to the Author):

NCOMMS-21-10551C

To have a clear perspective, I read again the manuscript before reading the rebuttal. Araujo et al. present an approach to improve the accuracy of species adsorbed on metal surfaces, aiming for its use in heterogeneous and electrocatalysis. It consists of taking a structure preconverged with a GGA (or MGGA: SCAN) exchange and correlation (xc) density functional; then selecting a cluster with and without the adsorbate and applying a hybrid calculation on both clusters. Upon a series of revisions, the approach is now presented as robust and well-converged. This is, it works well independently of the (M)GGA xc functional used, and the convergence with respect to the cluster size is also illustrated. The correction has perturbations in the order of 2-3 kcal mol⁻¹, particularly for small cluster sizes (<13 metal atoms). This is larger the chemical accuracy <1 kcal mol⁻¹, but it is still acceptable for applications in heterogeneous and electrocatalysis. In any case, the results obtained from the approach are more accurate than only using GGA/MGGA density functionals. Additional discussion about the underlying physical phenomena is also provided. As such, the science behind the Manuscript is good, although not fully novel.

Withal, the message is quite abstract. This by itself would not be problematic. But after three rounds of revision, there are still key descriptions that are unnecessarily obscure, unclear, or ambiguous (**). These are not simply typos or format issues, which still abundantly exist. The obscurity that still permeates the paper, even in key descriptions, will likely prevent attempts to reproduce the work and lower the interest of the diverse readership of Nature Communications. More importantly, the few physical insights that were discussed are not fully novel and may interest only a specialized readership. Reviewer #2 even used the term “unenthusiastic”.

Given that the approach is robust, but is poorly presented, the moderate-low novelty in the physical insights, the potentially moderate-low interest of broad readership, and the potential appeal for specialized readers, I recommend transferring the Manuscript to a more specialized Journal, such as Communication Chemistry or Communications Physics.

Issues that still obscure reproducibility:

** Lines 284-293: In this description, it is still not explicitly stated if Pt(111) and Pt(111)+O clusters had multiplicities of (a) 7-and-7, or (b) 9-and-7. The explanation given in lines 166-167 and 621-623 seems to point out option (a). Yet, lines 284-293 point to option (b). Should the spin states be similar (option b, line 287), or equal (option a, “same” in line 203)?

** Line 183: ^Ads,PBC instead of ^Ads,PBE. (Part of the description of the main Equation).

* Lines 227-230: Are the k-point meshes centered around the Gamma-point? Monkhorst-Pack grids containing even numbers of divisions, in non-squared cells such as the Pt(111), do not have an evenly distributed sample of k-points in the irreducible Brillouin zone (Unless they are centered in the Gamma-point). The lateral size of the cell is also missing here because it can give an idea if the grid size for the reciprocal space is sufficient or not.

* The equation shown in Supplementary Figure S1 seems similar but not equal to Equation 1 in the main text. Eg, the term “Ads” is missing. Do they point to different approaches?

Other changes:

** In the “Reporting Summary”, it was marked that “A full description of the statistical parameters including central tendency ... regression coefficient ... standard deviation ... uncertainties” does not apply. Actually, it does apply here. For instance, in Supplementary Figure S3, the regression errors

can be included, eg, $(1.0163 \pm \text{error1}) * x + (1.2201 \pm \text{error2})$. Most programs capable of doing regression can also report standard deviation for regression parameters: Microsoft Excel, LibreOffice Calc, Origin, Python...

* Line 156: (in Pt(111))

* Figure 3, vertical axis: "corretion".

* Figure 4, Figure 5, Supplementary Figure S2, Supplementary tables S1-S4, list of Experimental uncertainties in the Supplementary Information: Subscripts in chemical formulae to be included

* Supplementary Figure S1: exchange-correlation functionals are normally written in uppercase (eg, B3LYP instead of b3lyp).

* The supplementary references need to be extensively reviewed: Eg, Supplementary Reference 13: Nerskov instead of Nørskov, and all the format.

* Reference 25, 68, 69 in the main text: subscripts in the chemical formulae.

Suggestions:

* Even if the clusters, from the point of view of the molecule, locally resemble Pt(111) surfaces, calling them "Pt(111) cluster" complicates the understanding. To distinguish surfaces from clusters more clearly, surfaces can be called Pt(111), and clusters, eg, Pt₁₃ or Pt_n.

* Lines 52-56: These two sentences may be abbreviated to say their message more directly.

* Figure 3, caption, could indicate that Lines are guides to the eye.

* Figure 6: The bars plot can be changed to another with higher information density, such as a boxplot, and/or even putting directly the data points. That will also give an idea about how the errors are distributed, skewness, etc.

Reviewer #2 (Remarks to the Author):

The authors have addressed my first three comments. I also understand why point 4 is not easy to address retrospectively. In particular, the new figure 3 within the main manuscript is crucial. On a technical level, I have no further concerns and the manuscript merits publication.

Reviewer comments and our actions

Reviewer 1

To have a clear perspective, I read again the manuscript before reading the rebuttal. Araujo et al. present an approach to improve the accuracy of species adsorbed on metal surfaces, aiming for its use in heterogeneous and electrocatalysis. It consists of taking a structure preconverged with a GGA (or MGGA: SCAN) exchange and correlation (xc) density functional; then selecting a cluster with and without the adsorbate and applying a hybrid calculation on both clusters. Upon a series of revisions, the approach is now presented as robust and well-converged. This is, it works well independently of the (M)GGA xc functional used, and the convergence with respect to the cluster size is also illustrated. The correction has perturbations in the order of 2-3 kcal mol⁻¹, particularly for small cluster sizes (<13 metal atoms). This is larger than the chemical accuracy <1 kcal mol⁻¹, but it is still acceptable for applications in heterogeneous and electrocatalysis. In any case, the results obtained from the approach are more accurate than only using GGA/MGGA density functionals. Additional discussion about the underlying physical phenomena is also provided. As such, the science behind the Manuscript is good, although not fully novel.

Withal, the message is quite abstract. This by itself would not be problematic. But after three rounds of revision, there are still key descriptions that are unnecessarily obscure, unclear, or ambiguous (**). These are not simply typos or format issues, which still abundantly exist. The obscurity that still permeates the paper, even in key descriptions, will likely prevent attempts to reproduce the work and lower the interest of the diverse readership of Nature Communications. More importantly, the few physical insights that were discussed are not fully novel and may interest only a specialized readership. Reviewer #2 even used the term “unenthusiastic”.

Given that the approach is robust, but is poorly presented, the moderate-low novelty in the physical insights, the potentially moderate-low interest of broad readership, and the potential appeal for specialized readers, I recommend transferring the Manuscript to a more specialized Journal, such as Communication Chemistry or Communications Physics.

Issues that still obscure reproducibility:

Comment: Lines 284-293: In this description, it is still not explicitly stated if Pt(111) and Pt(111)+O clusters had multiplicities of (a) 7-and-7, or (b) 9-and-7. The explanation given in lines 166-167 and 621-623 seems to point out option (a). Yet, lines 284-293 point to option (b). Should the spin states be similar (option b, line 287), or equal (option a, “same” in line 203)?

Response: We have rewritten to better emphasize the difference between the chemisorption on the cluster (at the PBE+D3 level) and the use of the cluster for the correction:

Lines 165-167: *“We note, finally, that, since the aim here is only to correct the local bond using the cluster calculations, it is sufficient to ascertain **that the same electronic (and therefore spin) states are used both in the higher-level and simpler DFT calculations on the cluster.**”*

Lines 286-288: *“Subsequently, the clusters with adsorbate were allowed to vary their spin state by small amounts to find the lowest energy state; for the correction it is essential that the higher-level calculation also uses these same spin-states.”*

We do not see a problem with lines 621-623 (actually 630-632), but repeat them here: *“The constraint on the cluster calculation is that the electronic structure (e.g., spin state and spin-coupling within the cluster) should be the same for both the lower- and higher-level calculations.”*

Comment: Line 183: ^Ads,PBC instead of ^Ads,PBE. (Part of the description of the main Equation).

Response: Corrected.

Comment: Lines 227-230: Are the k-point meshes centered around the Gamma-point? Monkhorst-Pack grids containing even numbers of divisions, in non-squared cells such as the Pt(111), do not have an evenly distributed sample of k-points in the irreducible Brillouin zone (Unless they are centered in the Gamma-point). The lateral size of the cell is also missing here because it can give an idea if the grid size for the reciprocal space is sufficient or not.

Response: We have followed previous work where the convergence with k-point sampling was investigated and the present k-point grid was shown to give converged results. We have added the following text:

“At the optimized geometry, the self-consistent energies were recomputed with a cutoff energy of 700 eV and an (8x8x1) k-point mesh which has been shown to give converged results in earlier work.^{4,9}”

Comment: The equation shown in Supplementary Figure S1 seems similar but not equal to Equation 1 in the main text. Eg, the term “Ads” is missing. Do they point to different approaches?

Response: We thank the referee for pointing this out. It is now fixed in the supplementary info.

Other changes:

Comment: In the “Reporting Summary”, it was marked that “A full description of the statistical parameters including central tendency ... regression coefficient ... standard deviation ... uncertainties” does not apply. Actually, it does apply here. For instance, in Supplementary Figure S3, the regression errors can be included, eg, $(1.0163 \pm \text{error1}) * x + (1.2201 \pm \text{error2})$. Most programs capable of doing regression can also report standard deviation for regression parameters: Microsoft Excel, LibreOffice Calc, Origin, Python...

Response: The requested error estimates have been added in Figure S3.

Comment: Line 156: (in Pt(111))

Response: We have added the missing closing parenthesis.

Comment: Figure 3, vertical axis: "corretion".

Response: Corrected.

Comment: Figure 4, Figure 5, Supplementary Figure S2, Supplementary tables S1-S4, list of Experimental uncertainties in the Supplementary Information: Subscripts in chemical formulae to be included

Response: We have updated all chemical formulae to use subscripts.

Comment: Supplementary Figure S1: exchange-correlation functionals are normally written in uppercase (eg, B3LYP instead of b3lyp).

Response: Corrected.

Comment: The supplementary references need to be extensively reviewed: Eg, Supplementary Reference 13: Nerskov instead of Nørskov, and all the format.

Response: Corrected.

Comment: Reference 25, 68, 69 in the main text: subscripts in the chemical formulae.

Response: Corrected.

Suggestions:

Comment: Even if the clusters, from the point of view of the molecule, locally resemble Pt(111) surfaces, calling them "Pt(111) cluster" complicates the understanding. To distinguish surfaces from clusters more clearly, surfaces can be called Pt(111), and clusters, eg, Pt_13 or Pt_n.

Response: We prefer to keep the connection to the surface that the cluster represents. We have added a statement to this effect (page 8):

"Finally, to emphasize the connection to the surface from which each cluster is derived, we will in the following denote each cluster as the original surface, e.g., Pt(111) cluster rather than Pt₁₀; the individual clusters are shown in the Supplementary Information."

Comment: Lines 52-56: These two sentences may be abbreviated to say their message more directly.

Response: We have simplified to read:

"An adaptively weighted sum (SW-R88) of energies from RPBE and optB88-vdW has been proposed, aiming to properly describe both covalent and non-covalent interactions, and shown to result in errors smaller than BEEF-vdW, for instance.⁸"

Comment: Figure 3, caption, could indicate that Lines are guides to the eye.

Response: Done.

Comment: Figure 6: The bars plot can be changed to another with higher information density, such as a boxplot, and/or even putting directly the data points. That will also give an idea about how the errors are distributed, skewness, etc.

Response: We thank the referee for this suggestion. However, we believe the bar plot shows well the case by case error of our approach compared to PBE+D3 and also gives an overview of the main problematic reactions.